# ADVERSENESS VS. EQUILIBRIUM: EXPLORING GRAPH ADVERSARIAL RESILIENCE THROUGH DYNAMIC EQUILIBRIUM

## ABSTRACT

Adversarial attacks to graph analytics are gaining increased attention. To date, two lines of countermeasures have been proposed to resist various graph adversarial attacks from the perspectives of either graph per se or graph neural networks. Nevertheless, a fundamental question lies in whether there exists an intrinsic adversarial resilience state within a graph regime and how to find out such a critical state if exists. This paper contributes to tackle the above research questions from three unique perspectives: i) we regard the process of adversarial learning on graph as a complex multi-object dynamic system, and model the behavior of adversarial attack; ii) we propose a generalized theoretical framework to show the existence of critical adversarial resilience state; and iii) we develop a condensed one-dimensional function to capture the dynamic variation of graph regime under perturbations, and pinpoint the critical state through solving the equilibrium point of dynamic system. Multi-facet experiments are conducted to show that the proposed approach can significantly outperform the state-of-the-art defense methods under five commonly-used real-world datasets and several representative attacks.

## 1 INTRODUCTION

Graph adversarial attacks (GAAs) are launched through executing subtle perturbations on edges, weights and even features to the original clean graph (Mu et al., 2021), such as injecting/removing edges/nodes, or even modifying features/weights. To date, two main categories of GAAs emerges, one is injection attack (Zheng et al., 2020; Zou et al., 2021; Zhao & Zhang, 2025), i.e. inject new nodes into the original graph rather than straightly revise the existing characteristics. The other is modification attack (Zügner & Günnemann, 2019; Sun et al., 2020; Alom et al., 2025), i.e. directly change the original graph in terms of edges/nodes and/or features/weights.

Accordingly, to defend against GAAs, a set of adversarial defense methods are proposed from two perspectives: graph per se and graph neural networks. For the former, preprocessing countermeasures are utilized, for instance, GCN-SVD (Entezari et al., 2020) utilizes singular value decomposition (SVD) to decompose the adjacency matrix of perturbed graph and obtains a low-rank approximation matrix with the purpose of cleaning perturbation. GCN-Jaccard (Wu et al., 2019) removes those edges that connected nodes have low Jaccard similarity of features. Analogously, GNN-GUARD (Zhang & Zitnik, 2020) assigns higher weights to edges over similar nodes, but prunes edges over dissimilar nodes. Moreover, there also exist optimization countermeasures, e.g., Pro-GNN (Jin et al., 2020) treats adjacency matrix as learning parameters to implement optimization manipulation under the requirements of being close to original adjacency matrix, low-rank of learned adjacency matrix and feature smoothness. Referring to the variance-based attention, RGCN (Zhu et al., 2019) dispenses differential weights to distinct neighbors during convolution. Additionally, towards the inductive graph-learning tasks, Wang et al. (2023) propose secure graph-learning mechanism in terms of graph partitioning with fairness and balance. Li et al. (2024) propose adversarial training paradigm to resist graph poisoning attack from the out-of-distribution (OOD) perspective. These methods implement adversarial defense by handling graph structure, i.e. graph per se.

For the latter, the neural ordinary differential equation (ODE) (Chen et al., 2018) and partial differential equation (PDE) (Chamberlain et al., 2021b;a) have been attempted to defend against GAAs.

Song et al. (2022) state that graph PDE can validly resist topology perturbation by heat-diffusion model on a general Riemannian manifold. Zhao et al. (2023) propose an energy-conservation graph Hamiltonian flow to promote adversarial resilience. This line of research tries to adopt diffusion models in physics to conduct graph neural flow w.r.t. the attack resilience, and generates time-evolved embedding vectors for graph nodes, then utilizes backpropagation to minimize loss function for the parameters optimization.

Upon the two lines of studies above, we can highlight that: i) GAAs are usually launched through node/edge revision and/or weight/feature modification; ii) towards the graph per se, current defense methods almost concentrate on low-rank approximation of adjacency matrix of the perturbed graph, deletion of edges over dissimilar nodes, smoothness of node features, bias of assigned weights; and iii) towards neural networks, ODE and PDE are employed to infer robust embedding vectors to represent nodes. Our work falls into the category of graph per se.

We believe the inherent property of graph regime is the most influential factor for graph robustness, only a proper graph structure is built, can the adversarial resilience of graph regime exist. Then, a fundamental research question appears, that is, whether there exists a critical state of adversarial resilience within each graph regime, if yes, how to find out it. To surround this basic question, we in theory explore a generalized **Equili**brium point-based adversarial **Res**ilience approach **EquiliRes** to resist various graph adversarial attacks.

**Main Contributions:** i) A hypothesis is put forward to bridge the connection between adversarial (deep) learning and complex dynamic system, by which GAAs can be modeled appropriately; ii) A generalized theoretic framework is proposed to showcase the existence of critical adversarial resilience state of graph regime, and explicitly exhibit in what prerequisites and in what perturbation domains the equilibrium point can be located; iii) A condensed one-dimensional function is developed to demonstrate the dynamic variation of graph regime under adversarial perturbations, from which we concretize the equilibrium-point trajectory with stability theory, upon which an attack-resilient graph structure is built; and iv) Multi-facet experiments are executed to validate the effectiveness and rationality of our proposed hypothesis using five real-world datasets.

## 2 THEORETICAL FRAMEWORK FOR GRAPH ADVERSARIAL RESILIENCE

### 2.1 ADVERSARIAL RESILIENCE HYPOTHESIS

We first utilize a vanilla physical phenomenon to present the intuitive understanding as shown in Fig. 1(a), wherein three balls settle on three positions in a curved orbit. The difference between position $P_1$ and position $P_3$ is that the former has no friction while the latter has. At first, the three balls are settled in still, i.e. mathematically $\frac{dp(t)}{dt}\Big|_{t=0}$ =0, it means the first-order derivative of displacement $p(t)$ is zero, that is to say, the balls would keep stationary as the time $t$ changes if no extra action exists, these three positions are what is called equilibrium points. Nevertheless, when a force $F$ acts on the three balls, for position $P_1$ the ball would move back-and-forth repeatedly in a bounded range; for position $P_2$ the ball would move away and cannot go back any more; for position $P_3$ the ball would first move back-and-forth and finally become still due to the action of friction. Inspired by this phenomenon, we inspect whether a graph regime also has such equilibrium point(s) representing its adversarial resilience state under different-level adversarial perturbations (attacks).

*Remark* 2.1. Observed from the perspective of complex dynamic systems, the three balls' states respectively correspond to **Stability**, **Non-Stability**, and **Asymptotic Stability** as shown in Fig. 1(b). Stability denotes the ball's trajectory varies along a boundary (e.g. between radius $\tau$ and radius $r$); Non-Stability indicates the ball's trajectory departs from the original position and moves towards infinity (e.g. exceed radius $r$); Asymptotic Stability implies the ball's trajectory first departs from the original position, and finally stops at the original position (e.g. the origin) even if it may experience a long-time motion. Note that for a dynamic system, there may exist multiple equilibrium points, also called fixed points, but it does not mean all of them are asymptotically stable.

As well known, GAA aims to breach graph's resilience by continuously perturbing graph structure (topology) or/and node feature. From the viewpoint of dynamic system, each time the perturbation

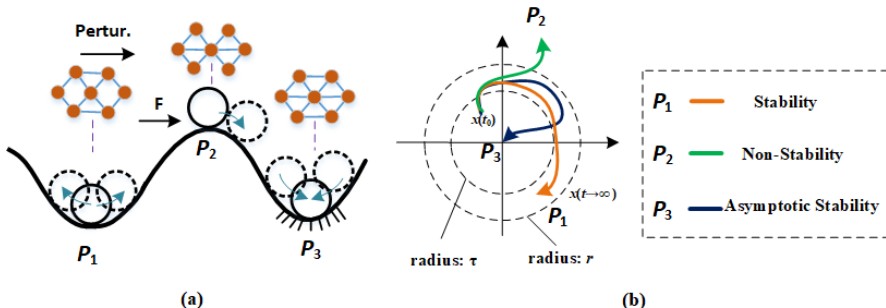

Figure 1: Three balls' motion states and equilibrium points.

tries to make graph deviate from the original stable state to non-stability state. Hence, we think each graph ought to have such an equilibrium point to keep robust, then propose the following hypothesis.

**Assumption 2.2. (Adversarial Resilience of Graph Regime)** For each graph regime, there exists an asymptotically-stable equilibrium point, corresponding to its critical adversarial resilience state, to keep it attack-resilient under continuous and bounded perturbations by adversarial attack, and such equilibrium point can be acquired as time goes.

GAAs generally deceive target model through dynamically adding/deleting important nodes/edges, in addition to modifying weights/features. Thus, we can abstractly mirror such continuous perturbations as the dynamic-variation process of complex multi-object physical systems, such as electro-magnetic field, charged particles, etc. As analyzed in Fig. 1(b), a dynamic system can fall into the three states: stable, non-stable, or asymptotically stable. Of course, staring from the stability state, we anticipate the dynamic system can finally converge into the asymptotically-stable state, this implies it can always go back to the original equilibrium point in the end even if a long-time dynamic undergoes. Therefore, referring to the converged trajectory of the ball at position $P_3$, we give the definition of asymptotic stability. The symbol "$\rightharpoonup$" denotes a vector representing the set of objects.

**Definition 2.3. (Asymptotic Stability (Khalil, 2015))** Let $\frac{d\vec{x}}{dt} = v(\vec{x})$ be a set of locally Lipschitz functions defined over a domain $\xi \subset \mathbb{R}^n$, and it contains the origin[1], i.e. $v(0)$=0. Then, the equilibrium point $x$=0 of $\frac{d\vec{x}}{dt}$ is asymptotically stable if there exists $\tau > 0$, s. t. $\left\|\vec{x}(t_0)\right\| < \tau \Rightarrow \lim_{t \to \infty} \vec{x}(t) = 0$.

*Remark* 2.4. For a dynamic system falling in the state of asymptotic stability, it means there exists asymptotically-stable equilibrium point and the system's dynamic variation trajectory will finally converges into such equilibrium point, no matter how long the dynamic may undergo. This procedure well-matches the behavior of adversarial attacks, that is to say, we can equivalently map the graph adversarial perturbation as dynamic system's oscillation. In view of this, regarding the adversarial perturbations on graph as dynamic variation of multi-object dynamic system is persuasive.

To make the asymptotically-stable equilibrium point (ASEP) fall into the origin, we give Lyapunov Criterion as stated in Appendix A. In the light of Lyapunov criterion, the origin must be an ASEP.

## 2.2 GAA MODELING

Graph dynamics (Kundu et al., 2022; Gao et al., 2016) are in general represented as a linear weighted average of the $N$ state variables each of which is associated with a node $i$, and the nonlinear influence of other nodes on node $i$. Hence, we can divide the adversarial perturbation on graph regime into two parts: i) linear perturbation, reflecting the graph is disturbed by a linear action; ii) nonlinear perturbation, implying the graph is disturbed by a nonlinear action. As time $t$ goes, the dynamic-

---

[1]The origin always can be obtained via coordinate transformation.

variation function under continuous perturbations can be defined as

$$\frac{d}{dt}\vec{x}(t) = A\vec{x}(t) + B\left(-\vec{\phi}(\cdot)\right), \tag{1}$$

where $\vec{x}(t)$ is an $N$-dimensional vector denoting the states of $N$ nodes, reflecting the linear perturbation, and $\phi(\cdot)$ is a nonlinear-mapping function representing the nonlinear perturbation. Equation 1 illuminates the dynamic variation of graph nodes under the continuous perturbations (edge removal/injection) by adversarial attack. Accordingly, each node may have a different states at different time $t$. It is also in concert with the training process of deep learning, namely, during each-round backpropagation-based parameter update, the pre-round output is adopted to compute gradient for the next-round parameter update. In the light of deep-learning principle, we here deem $\phi(\cdot)$ as the input to infer next-round dynamic update, and it should be presented by the output of pre-round perturbation result, in view of this, we can naturally define the output $\vec{y}(t) = C\vec{x}(t)$, which represents the pre-round perturbed states of $N$ nodes, thus we have $\phi(\cdot) = \phi\left(\vec{y}(t)\right)$ accordingly. Matrices $A$, $B$, $C$ are three general parameters, whose elements depend on the particular adversarial attack, the subsequent Equations 8-9 and associated parameters in Table 6 can interpret this point.

The integral from time $t_0$ to $t$ reflects a period of time aggregation of dynamic variation, which can be equivalently recognized as the process of adversarial learning epoch by epoch, thus the utilization of integral form is adequate to illuminate the effect of continuous adversarial perturbations on graph in a time frame. Therefore, we straightly define the time-aware aggregation as $\int_{t_0}^{t} x(\kappa)\,d\kappa$. Assume a persistent long-time adversarial learning process, i.e. $t \to \infty$, and simultaneously for the sake of derivative calculation, we introduce Laplace Transform for the above time-aware integral

$$L[x(t)] = X(s) = \int_{t_0=0}^{t\to\infty} x(t)e^{-st}dt, \tag{2}$$

where $s = \sigma + \omega j$, $j = \sqrt{-1}$ is a complex number. Given the property of linear transformation of Laplace Transform, we have $Y(s) = C \cdot X(s)$, and the first derivative's Laplace Transform is

$$L\left[\frac{d}{dt}x(t)\right] = \int_{t_0=0}^{t\to\infty}\left(\frac{d}{dt}x(t)\right)e^{-st}dt = \lim_{t\to\infty} x(t)e^{-st} - x(0) + sX(s). \tag{3}$$

In general, to achieve the convergence, the real part of complex number $s$ must satisfy $\text{Re}(s) > 0$, furthermore, the initial condition meets $x(0) = 0$, thus $L\left[\frac{d}{dt}x(t)\right] = sX(s)$. Substituting Equations 2 and 3 into Equation 1, an intermediate function $G(s)$ can be obtained between output and input,

$$\vec{X}(s) = (sI - A)^{-1} \cdot B \cdot L\left[\vec{\phi}\left(\vec{y}(t)\right)\right]$$

$$G(s) = \frac{\vec{Y}(s)}{L\left[\vec{\phi}\left(\vec{y}(t)\right)\right]} = \frac{C \cdot \vec{X}(s)}{L\left[\vec{\phi}\left(\vec{y}(t)\right)\right]} = C(sI - A)^{-1}B. \tag{4}$$

---

*Remark* 2.5. The function $G(s)$ reflects the intermediate variation process of information-passing from an input $L\left[\phi\left(\vec{y}\right)\right]$ to an output $Y(s)$ in Laplace form, thus, we can control the coefficients $\{A, B, C\}$ to obtain the result that we anticipate the system to output. From the viewpoint of adversarial resilience, we anticipate the graph finally converges into critical state of adversarial resilience (asymptotically-stable equilibrium point) to remain attack resilient, no mater how long it takes. In fact, you can image function $G(s)$ represents the intermediate process of adversarial learning.

---

We propose the following theorem to warrant Equation 1 converges into the ASEP.

**Theorem 2.6.** *(Existence of Asymptotically-Stable Equilibrium point) The dynamic-variance function (Equation 1) has asymptotically-stable equilibrium point(s) if matrix $A$ is Hurwitz, the output $\vec{y}(t)$ and its associated nonlinear-mapping input $\phi\left(\vec{y}(t)\right)$ satisfy the condition that $\vec{\phi}\left(\vec{y}(t)\right)^{T} \cdot \left[\vec{y}(t) - M\vec{\phi}\left(\vec{y}(t)\right)\right] > 0$, where matrix $M = diag\left(\frac{1}{k_1}, \cdots, \frac{1}{k_p}\right)$, $k_i > 0$. Furthermore, given $\psi = diag(\gamma_1, \cdots, \gamma_p)$, there exists constant $\gamma_i \geq 0$, such that $(1 + \lambda_k\gamma_i) \neq 0$ for each eigenvalue $\lambda_k$ of matrix $A$, and meanwhile $M + (I + s\psi)G(s)$ is strictly positive real.*

*Proof.* See Appendix B

## 3 ADVERSARIAL RESILIENCE INFERENCE

### 3.1 ONE-DIMENSIONAL PERTURBATION MAPPING

The work Kundu et al. (2022) studies single-node autonomous behavior (SNAB) and pairwise-node interaction behavior (PNIB), taking as a referral, our mentioned adversarial perturbation is investigated along this thought. In detail, we refer to the dynamic variation of graph under adversarial attack as coupled linear function $\chi\left(\vec{x}\right)$ and nonlinear function $\varphi\left(\vec{x},\vec{x}\right)$. Thus, for node $i$, we have

$$\frac{dx_i}{dt} = \tilde{A}_{(i)}\chi\left(x_i\right) + \sum_{j=1}^{N} \tilde{B}_{(ij)}\varphi(x_i, x_j), \tag{5}$$

where $\tilde{A}_{(i)}$ denotes the linear SNAB-caused dynamic-variation coefficient of node $i$, and $\tilde{B}_{(ij)}$ stands for the nonlinear PNIB-caused dynamic-variation coefficient.

Currently, the node is in general represented in a high-dimension space, to achieve lightweight computation, some dimension-reduction methods are adopted to accomplish low-dimension mapping. Considering the complexity of linkage/interactive relationship over nodes, we here employ one-dimensional condensed method to map the whole graph's dynamic variation. Resorting to Heterogeneous Mean-Field (HMF) approximation theory (Kundu et al., 2022), pursuant to the deduction step by step in Appendix E, we obtain the one-dimensional perturbation mapping function

$$\frac{d\tilde{x}}{dt} = \tilde{A}_{(\cdot)}\chi\left(\tilde{x}\right) + \tilde{\beta}\varphi\left(\tilde{x},\tilde{x}\right). \tag{6}$$

From Equation 6, we know the mapping function of the whole graph's dynamics is condensed as a single variable $\tilde{\beta}$, which can also be regarded as the reflection of robustness of the graph as a whole.

> *Remark* 3.1. The condensed one-dimensional mapping function may lead to somewhat loss of graph-structure information. Alternatively, the sophisticated two-dimensional or multi-dimensional mapping functions are workable as well. The principle to develop an appropriate mapping function lies in the capability of faithfully reflecting the caused dynamics/perturbations to the most extent.

### 3.2 EQUILIBRIUM-POINT EXPLORATION

Given that $\chi\left(x_i\right)$ is linear and $\varphi(x_i, x_j)$ is nonlinear, we can simply and generally define a dynamic-variation equation to concretize the process of adversarial perturbations as,

$$\frac{dx_i}{dt} = -\tilde{A}_{(i)}x_i + \sum_{j=1}^{N} \tilde{B}_{(ij)}\frac{Hx_j^2}{\theta x_j^2 + 1}. \tag{7}$$

The first term on the right hand of Equation 7 presents node $i$'s self-dynamics, for detail, we utilize the degree centrality of node $i$ to reflect the probability of being attacked. The second stands for the interactively-affected perturbation by its neighboring nodes. This simple yet effective one-dimensional function is developed to map the dynamics of graph regime as a whole, in fact, two-dimensional mapping or multi-dimensional mapping can be explored as well for distinct graph topologies, since our proposed asymptotically-stable theoretic framework fully supports multi-dimensional dynamic-variation function. The more accurate the modeling on GAAs, the higher the adversarial resilience would be. To get an accurate function to capture the dynamics, usually, some extra variable $\theta > 0$ is needed, and $H$ denotes the average in-degree weight of node $i$, implying the affected extent of perturbation by neighbors. Referring to Equation 6 in the condensed one-dimensional form, we can concretize Equation 7 as

$$\frac{d\tilde{x}}{dt} = -a\tilde{x} + \tilde{\beta}\frac{H\tilde{x}^2}{\theta\tilde{x}^2 + 1}, \tag{8}$$

where $a$ is a successor constant condensed from vector $\tilde{A}_{(i)}$. The equation above exhibits the detailed settings regrading the concrete graph attributes: degree centrality, in-degree weight, etc. Here what we need emphasize is that Equation 6 is a theoretical guidance, Equations 7 and 8 are concretized deductions w.r.t. specified graph topology. From Equation 8, we infer the following theorems.

**Theorem 3.2.** *Upon the condensed one-dimensional mapping function, a graph regime disturbed by graph adversarial attack can be finally converged into an asymptotically-stable equilibrium point if the constant $k$ in Theorem 2.6 belongs to the interval $\left[ H \cdot \left( 2\sqrt{\theta} \right)^{-1}, \infty \right)$.*

*Proof.* See Appendix F.

Theorem 3.2 gives the perturbation domain only within which the asymptotically-stable equilibrium point can be finally found out, and the following Theorem 3.3 warrants the existence of such point.

**Theorem 3.3.** *The condensed one-dimensional dynamic-variation function (Equation 8) is asymptotically stable.*

*Proof.* See Appendix G.

---

*Remark* 3.4. **Theorem 3.2** and **Theorem 3.3** guarantee our proposed one-dimensional dynamic-variation function has asymptotically-stable equilibrium point, i.e. there exist a critical state of adversarial resilience to keep graph regime attack-resilient. The parameter $\omega \in \mathbb{R}$ in proof of Theorem 3.3 means the equilibrium point can be found out in the real number field, simultaneously with the constraint of perturbation scope of $k$. In other words, the existence of equilibrium point can be only warranted within a bounded domain. Once the perturbation jumps out of the constraint domain, the critical state of adversarial resilience would disappear.

---

Seen from Equation 8, we know $\frac{d\tilde{x}}{dt}=0$ if $\tilde{x} = 0$, which means the entire graph is completely broken with no connection between nodes (Equation 33), this is also a so-called equilibrium point, but not what we anticipate although this case indeed happens in reality. Thereby, we need calculate other equilibrium points divided by $\tilde{x}$, and at the same time regard $\tilde{x}$ as the variable of $\tilde{\beta}$,

$$\tilde{\beta}\left(\tilde{x}\right) = a \cdot \frac{\theta \tilde{x}^2 + 1}{H\tilde{x}}. \tag{9}$$

Pursuant to this function, we know a trajectory consisting of infinite equilibrium points can be drawn in theory. Since our work focuses on enhancing adversarial resilience from the angle of graph topology, i.e. resist GAAs from the radical graph structure. In order to endow appropriate variables for Equation 8, we choose the basic graph property–Degree Centrality, which reflects the "hub" role in the connectivity of entire graph, and defined as $C_{deg}(i) = \frac{|d_i|}{(N-1)}$ where $d_i$ denotes the set of neighbors of node $i$. Accordingly, we can separately obtain in-degree centrality $C_{Indeg}(i) = \frac{|d_{Inneib}(i)|}{(N-1)}$ where $d_{Inneib}(i)$ denotes the set of node $i$'s in-going neighbors, and out-degree centrality $C_{Outdeg}(i) = \frac{|d_{Outneib}(i)|}{(N-1)}$ where $d_{Outneib}(i)$ indicates the set of node $i$'s out-going neighbors. As mentioned previously, we set $a = \eta \cdot \frac{\sum_i C_{Indeg}(i) + \sum_i C_{Outdeg}(i)}{N}$, $H = \frac{\sum_i d_{Inneib}(i)}{N}$, and use the deduced Equation 33 in Appendix E to calculate $\tilde{x}$. The subsequent experiment showcases this basic and simple degree-related property plays an important role to boost adversarial resilience under the guidance of equilibrium-point trajectory.

### 3.3 QUANTIFYING PERTURBATION BOUND

Theoretically, the perturbed-bound with attack budget can be quantified as sketched in Appendix H.

## 4 IMPLEMENTATION

The asymptotically-stable equilibrium point is in accordance with the critical state of adversarial resilience for graph regime, thus we take such equilibrium point as the referral to build a new attack-resilient graph topology for perturbed graph. In detail, resorting to the adjacency matrix of undisturbed graph, we first compute $\tilde{\beta}$ using Equation 9 to obtain an asymptotically-stable equilibrium point as shown "Original" in Fig. 2-6 in Appendix J.2. Section 5.2 will detail the computation of equilibrium-point trajectory. Then, we set up an iterative-optimization procedure to generate a

robust adjacency matrix referring to this asymptotically-stable equilibrium point, i.e., enforcing the modified adjacency matrix-based coordinator point to be close to the "Original" point. For easy understanding, we state the procedures in Algorithm 1 and Algorithm 2 in Appendix I.

## 5 EXPERIMENT EVALUATION

Apart from regular performance analytics, our experiments need first to answer three fundamental research questions: i) **RQ1:** Whether there is a correlation between adversarial learning and complex dynamic system; ii) **RQ2:** Whether Laplace Transfer is adaptable to embody the process of continuous adversarial perturbations; and iii) **RQ3:** Whether the condensed one-dimensional mapping function suffices to appropriately present the dynamics of entire graph regime.

### 5.1 CONFIGURATION

The configurations on the statistics of five commonly-used moderate- and large-scale datasets[2]: Polblogs, Cora, Cora_ML, Citeseer, and Amazon Photo, seven representative adversarial defense baselines: GCN (Kipf & Welling, 2017), GAT (Velickovic et al., 2018), RGCN (Zhu et al., 2019), GCN-SVD (Entezari et al., 2020), HANG (Zhao et al., 2023), Mid-GCN (Huang et al., 2023), and GOOD-AT (Li et al., 2024), three non-targeted attacks: Metattack (Zügner & Günnemann, 2019), CE-PGD (Xu et al., 2019), DICE (Waniek et al., 2016), and one targeted attack: GOttack (Alom et al., 2025), execution settings are detailed in Appendix J.1.

### 5.2 EQUILIBRIUM-POINT TRAJECTORY

To well predict the adverseness of the three GAAs, we use the parameters in Table 6 in Appendix J.2 to draw equilibrium-point trajectories with Polblogs, Cora_ML, Cora, Citeseer, and Amazon Photo in Fig. 2-6 in Appendix J.2, from which, we observe the adversarial resilience degrades almost along the trajectory curves, this phenomenon unveils three-aspect hints: i) the concordance of equilibrium-point trajectory and adverseness of adversarial attacks discloses the existence of correlation between adversarial learning and complex dynamic system, which answers the first question **RQ1**; ii) note that for each three curves drawn from the five datasets, the variation tendency of inferred adversarial resilience under different-level adversarial perturbations ranging from 3% to 30% goes exactly along the theoretical equilibrium-point curve which is deduced by Laplace Transfer, this unveils the rationality of introduction of Laplace Transfer for the modeling on graph adversarial attacks, which replies the second question **RQ2**; and iii) the condensed one-dimensional function-based equilibrium-point trajectory is appropriate to capture the dynamics of graph regime as a whole under the continuous adversarial perturbations, which answers the third question **RQ3**.

### 5.3 PERFORMANCE ANALYTICS

#### 5.3.1 ADVERSARIAL RESILIENCE PROMOTION

**Asymptotically-Stable Equilibrium-Point Trajectory Conducts Attack-Resilient Topology.** The results in Table 1, Tables 7-8 in Appendix J.3 showcase the remarkable robustness improvement of our EquiliRes compared to GCN-SVD and GCN. The adversarial perturbation rate (APR) 0% means there does not exist adversarial attack, we execute matrix-modification operation with a little percentage 1%, from which, we can deduce two-facet opinions: i) the accuracy of node classification is high without adversarial perturbation, e.g. approximately 94% with the three methods on Polblogs dataset; and ii) our EquiliRes gains larger accuracy compared to the two baselines, which implies our proposed asymptotic stability theory can validly locate the critical state of adversarial resilience to keep the graph more robust. Even for clean graph, our method still can further boost it.

For Metattack attack, the accuracy raises by 15.40%, 13.96%, 6.97% than the second best when the APRs are 15%, 20% and 25% on Polblogs, and by 5.67%, 6.51% and 4.93% on Citeseer when APRs are 10%, 15% and 20% respectively, which shows our proposed approach can effectively defend against such non-targeted gradient-based attack. For CE-PGD attack, although it perturbs clean graph in a maximum probability through enlarging the cross-entropy loss, pursuant to Theorem

---

[2]https://pytorch-geometric.readthedocs.io/en/latest/cheatsheet/data_cheatsheet.html

Table 1: Accuracy (%) of node classification on Polblogs (**bold**-the best)

| GAA | Per. Def. | 0% | 5% | 10% | 15% | 20% | 25% |
|---|---|---|---|---|---|---|---|
| Metattack | GCN | 94.26±1.24 | 76.35±1.39 | 70.61±1.51 | 67.66±1.39 | 67.45±1.45 | 66.34±1.54 |
| | GCN-SVD | 93.98±0.52 | **92.52±1.09** | 87.99±4.35 | 71.13±2.93 | 67.25±1.69 | 65.56±1.49 |
| | EquiliRes | **94.34±0.81** (0.08%↑) | 90.11±2.62 (2.41%↓) | **89.70±2.22** (1.71%↑) | **86.53±1.47** (15.40%↑) | **81.41±1.94** (13.96%↑) | **73.11±2.60** (6.97%↑) |
| CE-PGD | GCN | 94.26±1.24 | 84.07±0.65 | 79.19±1.08 | 76.86±0.91 | 75.37±1.05 | 74.47±1.14 |
| | GCN-SVD | 93.98±0.52 | 85.31±1.06 | 79.43±1.04 | 76.82±1.06 | 75.59±1.00 | 74.80±1.26 |
| | EquiliRes | **94.34±0.81** (0.08%↑) | **86.69±1.19** (1.38%↑) | **82.67±1.12** (3.24%↑) | **79.93±1.39** (3.07%↑) | **76.82±1.01** (1.23%↑) | **75.59±1.41** (0.79%↑) |
| DICE | GCN | 94.26±1.24 | 86.58±1.22 | 80.83±0.96 | 77.42±1.07 | 75.44±1.13 | 73.37±0.99 |
| | GCN-SVD | 93.98±0.52 | **90.83±0.78** | 88.29±0.50 | 85.56±0.66 | 82.92±1.20 | 81.29±0.96 |
| | EquiliRes | **94.34±0.81** (0.08%↑) | 89.47±0.95 (1.36%↓) | **88.43±2.32** (0.14%↑) | **86.85±2.11** (1.29%↑) | **84.75±1.54** (1.83%↑) | **83.45±1.63** (2.16%↑) |

Table 2: Accuracy (%) of node classification on Citeseer under Gottack

| GAA | RNS Def. | 0% | 10% | 30% | 50% | 70% | 90% |
|---|---|---|---|---|---|---|---|
| GOttack | GCN-SVD | 49.54±1.71 | 43.46±1.49 | 42.60±0.33 | 45.45±1.46 | 47.19±1.00 | 43.91±0.79 |
| | EquiliRes | **60.24±0.69** (10.70%↑) | **60.04±0.69** (16.58%↑) | **59.77±0.41** (17.17%↑) | **60.43±0.69** (14.98%↑) | **59.94±0.66** (12.75%↑) | **60.01±0.59** (16.10%↑) |

3.3, we know that, under a bounded perturbation, our proposed dynamic-variation function can finally ends up at asymptotically-stable equilibrium point to make the graph attack-resilient, thus our EquiliRes possesses 3.24%, 3.07% and 1.23% improvement on Polblogs, and 2.65%, 2.53% and 2.30% on Citeseer when APRs are 10%, 15% and 20% respectively.

Different from the three non-targeted attacks, GOttack is a targeted attack that aims to modify (add/delete) the influential edges to targeted nodes. We set the ratio of node selection (**RNS**) as 0%, 10%, 30%, 50%, 70%, 90%, from the experimental results in Table 2, we observe that EquiliRes significantly outperforms the analogous adjacency matrix-purified method GCN-SVD, this is because although GOttack aims to attack targeted nodes, it in fact performs perturbation through modifying edge, our topology-focused EquiliRes still can find out the critical attack-resilient state.

Table 3: Accuracy (%) of node classification under combination with EquiliRes using Metattack

| Dataset | Per. Def. | 0% | 5% | 10% | 15% | 20% | 25% |
|---|---|---|---|---|---|---|---|
| Polblogs | GAT | 94.88±0.42 | 86.93±2.39 | 76.89±2.14 | 71.78±1.46 | 68.29±1.83 | 66.99±1.64 |
| | Ours+GAT | **95.12±0.52** (0.24%↑) | **92.21±1.20** (5.28%↑) | **89.94±1.89** (13.05%↑) | **87.96±2.09** (16.18%↑) | **84.17±2.26** (15.88%↑) | **76.08±4.80** (9.09%↑) |
| | RGCN | 94.87±0.22 | 77.87±0.69 | 70.50±0.37 | 67.60±0.24 | 68.35±0.18 | 67.68±0.30 |
| | Ours+RGCN | **95.15±0.28** (0.28%↑) | **84.16±0.33** (6.29%↑) | **77.37±0.42** (6.87%↑) | **75.03±0.31** (7.43%↑) | **72.42±0.28** (4.07%↑) | **70.05±0.25** (2.37%↑) |
| | HANG | 93.72±1.26 | 88.88±1.83 | 87.45±1.21 | 86.53±1.47 | 81.41±1.94 | 73.11±2.60 |
| | Ours+HANG | **93.85±0.75** (0.13%↑) | **90.73±1.14** (1.85%↑) | **89.65±2.13** (2.20%↑) | **86.67±2.08** (0.14%↑) | **82.71±2.66** (1.30%↑) | **76.11±4.53** (3.00%↑) |
| | Mid-GCN | 91.11±1.82 | 89.44±3.35 | 84.71±4.94 | 72.59±4.10 | 67.16±1.38 | 64.70±1.08 |
| | Ours+Mid-GCN | **92.15±2.26** (1.04%↑) | **90.82±2.49** (1.38%↑) | **85.27±4.67** (0.56%↑) | **74.29±1.96** (1.70%↑) | **70.88±1.51** (3.72%↑) | **67.67±1.39** (2.97%↑) |
| | GOOD-AT | 95.17±0.21 | 94.78±0.52 | 94.97±0.34 | 93.50±0.98 | 93.68±1.36 | 93.68±1.36 |
| | Ours+GOOD-AT | **95.27±0.20** (0.10%↑) | **95.42±0.27** (0.64%↑) | **95.17±0.47** (0.20%↑) | **95.24±0.35** (1.74%↑) | **94.05±0.57** (0.37%↑) | **93.80±0.96** (0.12%↑) |

**Taking Equilibrium State of Graph Regime as Booster to Promote Adversarial Resilience.** Besides the adjacency matrix optimization, other robustness-enhancing mechanisms exist, like graph attention network, ODE-based graph neural flow, mid-frequency signals, etc. To furtherly explore

Table 4: Accuracy (%) of node classification with/without referring to ASEP on Polblogs

| GAA | Per. | 0% | 5% | 10% | 15% | 20% | 25% |
|---|---|---|---|---|---|---|---|
| Metattack | w/o | 93.15±0.26 | 78.87±1.08 | 72.39±1.25 | 69.24±0.74 | 69.53±0.99 | 68.08±0.30 |
| | w/ | **94.34±0.81** (**1.19%↑**) | **90.11±2.62** (**11.24%↑**) | **89.70±2.22** (**17.31%↑**) | **86.53±1.47** (**17.29%↑**) | **81.41±1.94** (**11.88%↑**) | **73.11±2.60** (**5.03%↑**) |
| CE-PGD | w/o | 93.15±0.26 | 84.14±0.15 | 79.01±0.14 | 77.02±0.11 | 75.43±0.08 | 73.74±0.07 |
| | w/ | **94.34±0.81** (**1.19%↑**) | **86.69±1.19** (**2.55%↑**) | **82.67±1.12** (**3.66%↑**) | **79.93±1.39** (**2.91%↑**) | **76.82±1.01** (**1.39%↑**) | **75.59±1.41** (**1.85%↑**) |
| DICE | w/o | 93.15±0.26 | 86.74±0.42 | 85.63±0.72 | 81.61±0.79 | 80.15±0.38 | 76.50±0.52 |
| | w/ | **94.34±0.81** (**1.19%↑**) | **89.47±0.95** (**2.73%↑**) | **88.43±2.32** (**2.80%↑**) | **86.85±2.11** (**5.24%↑**) | **84.75±1.54** (**4.60%↑**) | **83.45±1.63** (**6.95%↑**) |

the effectiveness of our work, we at first find out the asymptotically-stable equilibrium point and generate associated topology, then utilize the existing methods to execute node-classification task. Table 3 and Table 9 (Appendix J.3) state the results, wherein the sign "**+**" indicates our EquiliRes as the preliminary is first executed, then followed by GAT, RGCN, HANG, Mid-GCN, and GOOD-AT.

Overall, referring to our proposed EquiliRes as a booster can significantly enhance the performance of existing defense mechanisms, upon which the different-level advancement on the five datasets at APR 0% unveils that even a clean graph can be enhanced through constructing a stable/equilibrium-state graph topology at first. Moreover, GAT improves the accuracy by on-average 11.90% on Polblogs, 10.916% on Amazon Photo, 4.68% on Cora_ML, 2.02% on Cora, 2.58% on Citeseer, HANG raises by on-average 1.70%, 0.82%, 5.02%, 1.99%, 2.55%, Mid-GCN promotes by on-average 2.07%, 1.18%, 0.42%, 2.20%, 0.61%, as well as RGCN and GOOD-AT with enhancement.

### 5.3.2 ABLATION

We run ablation experiments with/without referring to ASEP as guidance on Polblogs, Cora_ML, and Cora. From the experimental results in Table 4 and Tables 10-11 in Appendix J.4, we can observe the performance referring ASEP as a guidance significantly outperforms that without it.

### 5.3.3 ANALYTICS ON RANK/SINGULAR VARIATION AND TIME OVERHEAD

GAA could enlarge the rank and singular values of adjacency matrix (Jin et al., 2020; Zhao et al., 2023), to resist that, our experiments showcase EquiliRes can indeed decease them when the graph topology matches the equilibrium-point state. The analytics are detailed in Appendices J.5 and J.6. At same time, the computation complexity and its scalability are given in Appendix J.7.

## 6 RELATED WORK AND OPEN ISSUES

We review the related work in Appendix K. Simultaneously, we also discuss the behind motivation why to bridge the connection between adversarial learning and dynamic systems, along with three open issues from the perspectives of condensed one-dimensional mapping function, adversarial resilience on multi-modal data, and parameters of resilience-declining function in Appendix L.

## 7 CONCLUSION

Upon our proposed hypothesis that each graph regime has a critical state to warrant its adversarial resilience, then, resort to the stability theory to find out asymptotically-stable equilibrium point corresponding to the graph's critical state of adversarial resilience. Referring to such equilibrium-point trajectory, we iteratively modify the adjacency matrix to infer an attack-resilient topology. Multi-facet experiments validate the effectiveness and the rationality of our hypothesis. Compared to SOTA methods, our work has a remarkable improvement on the robustness. To us, the most important innovation lies in bridging the connection between adversarial learning and complex dynamic system, and providing a provable solution to pinpoint the critical state of adversarial resilience. We believe our work will stimulate more exploration on the intrinsic robustness of graph in future.

## REPRODUCIBILITY STATEMENT

In our work, all the experiments of ours and baselines are performed in an open-source benchmark for graph analytics on adversarial attack and defense, thus, the experimental results are reproducible. We publicize our source code at Github to enhance sharing on our scientific findings.

**Data Collection and Use.**

The five datasets used in our experiments, i.e. Polblogs, Cora_ML, Cora, Citeseer, and Amazon Photo are all open-sourced and commonly-used in academic domain. They all can be downloaded directly from Internet:`http://www.cs.umd.edu/~sen/lbc-proj/LBC.html`.

**Source Code.**

We open-source the code at Github: `https://github.com/Anony4OpenScience/EquiliRes_V1.0`.

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

## A  LYAPUNOV CRITERION

In the domain of system stability, Lyapunov criterion as a classic methodology provides the strict condition to guarantee the origin must be an asymptotically-stable equilibrium point for multi-object dynamic systems. For the detailed proof, refer to the reference Khalil (2015).

**Theorem A.1.** *(Lyapunov Criterion (Khalil, 2015)) Let $v(\vec{x})$ be a locally Lipschitz function over a domain $\xi \subset \mathbb{R}^n$, which contains the origin. Define a continuously differentiable function $V\left(\vec{x}\right)$ over domain $\xi$, such that*

$$V\left(0\right) = 0, V\left(\vec{x}\right) > 0 \ \ for \ all \ \ \vec{x} \in \xi \ \ and \ \ \vec{x} \neq 0$$
$$\frac{d}{dt}V\left(\vec{x}\right) < 0 \ \ for \ \ all \ \ \vec{x} \in \xi \ \ and \ \ \vec{x} \neq 0 \tag{10}.$$

This Lyapunov criterion gives a general methodology to prove whether an asymptotically-stable equilibrium point exists or not, upon which, we infer the subsequent theorems in this paper.

## B  PROOF OF THEOREM 2.6

Pursuant to Theorem A.1, a candidate Lyapunov function is defined at first, then the quadratic-form is employed for the linear-part perturbation. For the conciseness, variable $t$ is omitted hereafter,

$$V_{Linear}\left(\vec{x}\right) = \frac{1}{2}\vec{x}^T P \vec{x}. \tag{11}$$

For the nonlinear-part perturbation, a reasonable and valid manner is to use the integral form to present the aggregated effect of continuous perturbations,

$$V_{Nonlinear}\left(\vec{x}\right) = \sum_{i=1}^{m} \gamma_i \int_{0}^{y_i} \phi_i(\sigma)d\sigma. \tag{12}$$

According to Theorem A.1 and Equation 1, we know $V_{Linear}$ ought to be positive definite, this implies the matrix must be $P > 0$ and symmetric. The first-order derivative of linear-part perturbation and nonlinear-part perturbation can be respectively calculated as:

$$\frac{d}{dt}V_{Linear}\left(\vec{x}\right) = \frac{1}{2}\vec{x}^T\left(PA + A^T P\right)\vec{x} + \vec{x}^T PB\left(-\phi\left(\vec{y}\right)\right). \tag{13}$$

$$\frac{d}{dt}V_{Nonlinear}\left(\vec{x}\right) = \phi^T\left(\vec{y}\right)\psi C\left(A\vec{x} + B\left(-\phi\left(\vec{y}\right)\right)\right). \tag{14}$$

Referring to the aforementioned Lyapunov Criterion, $(PA + A^T P)$ is anticipated to be negative definite, to prove this point, the following theorem is provided.

**Theorem B.1.** *For any given positive definite and symmetric matrix $\tilde{N}$, and its decomposed non-singular matrix $\hat{N}$, $PA + A^T P = -\hat{N}^T \hat{N}$ has a unique positive definite and symmetric solution $P$ if all eigenvalues of matrix $A$ have negative-real parts.*

*Proof.* See Appendix C.

Theorem B.1 guarantees the negative definite of first part of Equation 13. Next, turn to the rest part $\vec{x}^T PB\left(-\phi\left(\vec{y}\right)\right)$ and $\frac{d}{dt}V_{Nonlinear}\left(\vec{x}\right)$. Similarly, we also attempt to match a quadratic form,

$$\begin{aligned}
\frac{d}{dt}V\left(\vec{x}\right) &= \frac{1}{2}\vec{x}^T\left(-\hat{N}^T \hat{N}\right)\vec{x} + \vec{x}^T PB\left(-\phi\left(\vec{y}\right)\right) \\
&\quad + \phi\left(\vec{y}\right)^T \psi C\left(A\vec{x} + B\left(-\phi\left(\vec{y}\right)\right)\right) \\
&= -\frac{1}{2}\left(\hat{N}\vec{x} - W\phi\left(\vec{y}\right)\right)^T\left(\hat{N}\vec{x} - W\phi\left(\vec{y}\right)\right) \\
&\quad - \phi\left(\vec{y}\right)^T\left[\vec{y} - M\phi\left(\vec{y}\right)\right]
\end{aligned} \tag{15}$$

On account of the negative semidefinite of $\frac{d}{dt}V\left(\vec{x}\right)$, we deduce the following theorem.

**Theorem B.2.** *There exist a diagonal matrix $M$, such that the perturbation output $\vec{y}$ and its associated nonlinear-mapping input $\phi\left(\vec{y}\right)$ satisfying $\vec{\phi}\left(y\right)^T \cdot \left[\vec{y} - M\vec{\phi}\left(y\right)\right] \geq 0$, a matrix $W$ and a diagonal matrix $\psi$, such that $PB = -\hat{N}^T W + A^T C^T \psi + C^T$, $W^T W = \psi CB + 2M + B^T C^T \psi$.*

*Proof.* See Appendix D.

Theorem B.1 and Theorem B.2 warrant the realization of Equation 15, namely the first-order derivative of candidate Lyapunov function is negative semidefinite. To finally warrant it converges into the asymptotically-stable equilibrium point, we anticipate it to be negative definite categorically,

$$\frac{d}{dt}V\left(\vec{x}\right) < -\frac{1}{2}\varepsilon\vec{x}^T P\vec{x}, \tag{16}$$

where $\varepsilon$ is an arbitrarily small positive number. To achieve this requirement, Equation 15 must satisfy $PA + A^T P = -\hat{N}^T \hat{N} - \varepsilon P$. Now, we need prove the existence of such an $\varepsilon$. According to

Theorem B.1, we know if $(PA + A^T P)$ is positive definite and symmetric, then it can be equal to the multiplication of a nonsingular matrix by its transpose. That is to say, we need prove $\hat{N}^T \hat{N} + \varepsilon P$ is positive definite (obviously symmetric), thus we have

$$\vec{x}^T \hat{N}^T \hat{N} \vec{x} + \varepsilon \int_0^t \vec{x}^T e^{\left(A^T \sigma\right)} \hat{N}^T \hat{N} e^{(A\sigma)} \vec{x} d\sigma > 0$$
$$\varepsilon > -\frac{\left\| \vec{x} \hat{N} \right\|_2^2}{\int_{t_0}^t \left\| \vec{x} e^{(A\sigma)} \hat{N} \right\|_2^2 d\sigma} \tag{17}$$

Visibly, there indeed exists such a positive number $\varepsilon \in (0, \infty)$. Of course, the interval can be extended to $\left( - \left\| \vec{x} \hat{N} \right\|_2^2 \Big/ \int_{t_0}^t \left\| \vec{x} e^{(A\sigma)} \hat{N} \right\|_2^2 d\sigma, \infty \right)$, concretely, it is subject to the eigenvalues of matrix $A$.

## C  PROOF OF THEOREM B.1

At first, establish the connection to matrix $A$ with time variable $t$ and assume an integral form for matrix $P$,

$$P = \int_0^t e^{\left(A^T \sigma\right)} \tilde{N} e^{(A\sigma)} d\sigma. \tag{18}$$

Then, we have that

$$\begin{aligned}
&PA + A^T P \\
&= \int_0^t e^{\left(A^T \sigma\right)} \tilde{N} e^{(A\sigma)} A d\sigma + \int_0^t A^T e^{\left(A^T \sigma\right)} \tilde{N} e^{(A\sigma)} d\sigma \\
&= \int_0^t \frac{d}{dt} \left( e^{\left(A^T \sigma\right)} \tilde{N} e^{(A\sigma)} \right) d\sigma \\
&= e^{\left(A^T \sigma\right)} \tilde{N} e^{(A\sigma)} \Big|_0^t
\end{aligned} \tag{19}$$

Let $\lambda_i$ $(i=1, \cdots, n)$ be the eigenvalues of matrix $A$ with associated eigenvector $v_i$, thus $A$ can be diagonalized as: $\Lambda = \hat{M}^{-1} A \hat{M}$, where $\hat{M} = [v_1, v_2, \cdots, v_n], \Lambda = diag\{\lambda_1, \cdots, \lambda_n\}$.

Next, calculate $e^{(At)}$ using Taylor series,

$$\begin{aligned}
e^{(At)} &= I + tA + \frac{t^2}{2!} A^2 + \cdots = \sum_{k=0}^{\infty} \frac{1}{k!} t^k A^k \\
&= MM^{-1} + tM\Lambda M^{-1} + \frac{t^2}{2!} M\Lambda^2 M^{-1} + \cdots
\end{aligned} \tag{20}$$

thus,

$$\begin{aligned}
M^{-1} e^{(At)} M &= I + t\Lambda + \frac{t^2}{2!} \Lambda^2 + \cdots \\
&= \begin{pmatrix} e^{\lambda_1 t} & & 0 \\ & \ddots & \\ 0 & & e^{\lambda_n t} \end{pmatrix} = e^{(\Lambda t)}
\end{aligned} \tag{21}$$

Since all the eigenvalues of $A$ have negative-real parts, thus we have that $e^{(At)} \to 0$ when $t \to \infty$, that is to say, through a long-time adversarial learning, $PA + A^T P$ will equal to $-\tilde{N}$. Due to the symmetry and positive-definite property of matrix $\tilde{N}$, it can be decomposed into the multiplication of one nonsingular matrix by its transpose[3], i.e. $\tilde{N} = \hat{N}^T \hat{N}$, which further indicates the symmetry of matrix $P$. Next, judge its definite property,

$$\begin{aligned}
x^T P x &= \int_0^t x^T e^{\left(A^T \sigma\right)} \hat{N}^T \hat{N} e^{(A\sigma)} x dt \\
&= \int_0^t \left\| x^T e^{\left(A^T \sigma\right)} \hat{N}^T \right\|_2^2 d\sigma > 0
\end{aligned} \tag{22}$$

Thus, there exists a unique positive definite solution $P$.

---

[3][Chen (1999), Theorem 3.7]

## D  PROOF OF THEOREM B.2

We first introduce a theorem[4], namely, for a $p \times p$ rational positive-real and Hurwitz[5] matrix $Z(s)$, there exists an $r \times p$ Hurwitz matrix $Q(s)$ such that

$$\tilde{Z}(s) = Z(s) + Z^T(-s) = Q^T(-s)Q(s), \tag{23}$$

where $\hat{r}$ is the rank of $Z(s) + Z^T(-s)$ over the field of rational functions of $s$, and rank $Q(s) = \hat{r}$ for $Res[s] > 0$.

To prove the Theorem B.2, we need extend $\vec{y} = C\vec{x} + D\vec{u}$, where $\vec{u}$ can be seen as a flexible adjustment to the output $\vec{y}$ by design, it can be degraded to the original if $D=0$.

Let $\{\bar{A}, \bar{B}, \bar{C}, \bar{D}\}$ be a minimal realization[6] of positive-real function matrix $Z(s)$. Pursuant to Equation 23, there exists an $\hat{r} \times p$ function matrix $Q(s)$ satisfying it, and let $\{F, G, L, J\}$ be a minimal realization of $Q(s)$, thus, the minimal realization of $Q^T(s)$ is $\{-F^T, L^T, -G^T, J^T\}$. Then, the realization of $Q^T(-s) \cdot Q(s)$ can be furtherly inferred as:

$$\left\{\hat{A}, \hat{B}, \hat{C}, \hat{D}\right\} = \left\{ \begin{pmatrix} F & 0 \\ L^T L & -F^T \end{pmatrix}, \begin{pmatrix} G \\ L^T J \end{pmatrix}, \left(J^T L, -G^T\right), J^T J \right\}. \tag{24}$$

Since both $Z(s)$ and $Z^T(-s)$ are Hurwitz, they have no common poles. Referring to the claim [(Duffin & Hazony, 1963), Theorem 7], we have $\delta\left[Z(s) + Z^T(-s)\right] = \delta[Z(s)] + \delta[Z^T(s)] = 2\delta[Z(s)]$, where $\delta[Z(\cdot)]$ denotes the degree of matrix $Z(\cdot)$. Therefore, $\left\{\hat{A}, \hat{B}, \hat{C}, \hat{D}\right\}$ is the minimal realization of $Q^T(-s)Q(s)$. From the claim [(Chen, 1999), Theorem 7.2], we know $\left(\hat{A}, \hat{B}\right)$ is controllable and $\left(\hat{A}, \hat{C}\right)$ is observable.

Given $(F, L)$ is observable and $F$ is Hurwitz, referring to Theorem B.1, there must exist a unique symmetric solution $K$, such that $KF + F^T K = -L^T L$. According to the proposition [(Ho & Kalman, 1966), Proposition 3], there must exist a nonsingular $T$ to achieve a similar transformation from a minimal realization $\left\{\hat{A}, \hat{B}, \hat{C}\hat{D}\right\}$ to another minimal realization $\left\{T\hat{A}T^{-1}, T\hat{B}, \hat{C}T^{-1}, \hat{D}\right\}$. Use $T = \begin{pmatrix} I & 0 \\ K & I \end{pmatrix}$ to make a new minimal realization,

$$\left\{\hat{A}_1, \hat{B}_1, \hat{C}_1, \hat{D}_1\right\} = \left\{ \begin{pmatrix} F & 0 \\ 0 & -F^T \end{pmatrix}, \begin{pmatrix} G \\ KG + L^T J \end{pmatrix}, \left(J^T L + G^T K, -G^T\right), J^T J \right\}. \tag{25}$$

Equivalently, we directly calculate the minimal realization of $Z(s) + Z^T(-s)$,

$$\left\{\hat{A}_2, \hat{B}_2, \hat{C}_2, \hat{D}_2\right\} = \left\{ \begin{pmatrix} \bar{A} & 0 \\ 0 & -\bar{A}^T \end{pmatrix}, \begin{pmatrix} \bar{B} \\ \bar{C}^T \end{pmatrix}, \left(\bar{C}, -\bar{B}^T\right), \bar{D} + \bar{D}^T \right\}. \tag{26}$$

Given the eigenvalues of matrix $A$ are all in the left-half plane (Hurwitz), and the eigenvalues of $-A^T$ is all in the right-half plane, indicating no common pole exists. Hence, we conclude the realization $\left\{\hat{A}_2, \hat{B}_2, \hat{C}_2, \hat{D}_2\right\}$ is minimal as well. Therefore, there must exist a nonsingular matrix $R$ such that $\left\{\hat{A}_1 = R\hat{A}_2 R^{-1}, \hat{B}_1 = R\hat{B}_2, \hat{C}_1 = \hat{C}_2 R^{-1}, \hat{D}_1 = \hat{D}_2\right\}$. Suppose $R$ is a block matrix as follows:

$$\begin{pmatrix} R_{11} & R_{12} \\ R_{21} & R_{22} \end{pmatrix}$$

---

[4][Youla (1961), Theorem 2]

[5]All the eigenvalues are strictly in the left half plane.

[6]For a given $Z(s)$, there exist many sets $\{\bar{A}, \bar{B}, \bar{C}, \bar{D}\}$ to realize, however, there must be a minimal-dimension of matrix $\bar{A}$, such a realization associated with minimal dimension $\bar{A}$ is termed a minimal realization.

Thus, we can deduce that $-\mathrm{R}_{12}\bar{\mathrm{A}}^{\mathrm{T}} = \mathrm{FR}_{12}$. Then premultiplying by $\exp(Ft)$ and postmultiplying by $\exp\left(A^T t\right)$, then we have

$$
\begin{aligned}
\frac{d}{dt}\left(\exp(Ft) R_{12} \exp(\bar{A}^T t)\right) &= \\
\exp(Ft)\left[FR_{12} + R_{12}\bar{A}^T\right]\exp(\bar{A}^T t) &= 0
\end{aligned}
\tag{27}
$$

Thus, $\exp(Ft) R_{12} \exp(\bar{A}^T t)$ is a constant for all $t$. Furthermore, $R_{12} = \exp(Ft) R_{12} \exp(\bar{A}^T t)$ at $t=0$, also $R_{12} = \exp(Ft) R_{12} \exp(\bar{A}^T t) \to 0$ at $t \to \infty$, thus $R_{12}=0$. The sequel equations can be inferred, i.e., $F = R_{11}\bar{A}R_{11}^{-1}$, $G = R_{11}\bar{B}$, $J^T L + G^T K = \bar{C}R_{11}^{-1}$, $R_{22}^T G = \bar{B}$.

Assume $P = R_{11}^T K R_{11}$, $\hat{N} = LR_{11}$, $W = J$, then we can infer that

$$
\begin{aligned}
P\bar{B} &= \bar{C}^T - \hat{N}^T W \\
W^T W &= \bar{D} + \bar{D}^T
\end{aligned}
\tag{28}
$$

On the basis of Equation 4, we redefine a new transfer function $\breve{G}(s)$ as,

$$
\begin{aligned}
\breve{G}(s) &= M + (I + s\psi) G(s) \\
&= M + C(sI - A)^{-1} B + \psi C(sI - A + A)(sI - A)^{-1} B \\
&= M + (C + \psi CA)(sI - A)^{-1} B + \psi CB
\end{aligned}
\tag{29}
$$

Let $\lambda_k$ be an eigenvalue of matrix $A$ and $v_k$ be the associated eigenvector. Then,

$$
(C + \psi CA) v_k = (C + \psi C\lambda_k) v_k = (I + \lambda_k \psi) Cv_k.
\tag{30}
$$

The condition implies $(A, C)$ is observable. Naturally, Equation 29 can be recognized as a quadri-tuple $\{A', B', C', D'\}$, namely $A' = A$, $B' = B$, $C' = (C + \psi CA)$, $D' = M + \psi CB$. If $\breve{G}(s)$ is strictly positive real and Hurwitz, referring to the conclusive Equation 28, we can easily obtain the results $PB = -\hat{N}^T W + A^T C^T \psi + C^T$, $W^T W = \psi CB + 2M + B^T C^T \psi$. That is to say, there exist a matrix $W$ and a diagonal matrix $\psi$ that make Equation 15 correct.

# E  DEDUCTION OF ONE-DIMENSIONAL MAPPING FUNCTION

Let $\Gamma^{in} = \left\{ \eta_i^{in} \big| \forall i \in [1, N], \eta_i^{in} = \sum_{j=1}^N \tilde{B}_{(ij)} \right\}$ and $\vartheta_j(x_i) = \varphi(x_i, x_j)$, where $\tilde{B}$ is the adjacency matrix, then

$$
\sum_{j=1}^N \tilde{B}_{(ij)} \varphi(x_i, x_j) = \eta_i^{in} \langle \vartheta_j(x_i) \rangle,
\tag{31}
$$

where $\langle \vartheta_j(x_i) \rangle$ denotes the average of PNIB-caused perturbation by node $i$'s neighbor $j$. Pursuant to Heterogeneous Mean-Field (HMF) approximation theory (Kundu et al., 2022), node $i$'s in-neighbor $j$ provides weight proportional to node $j$'s out-degree, then we have the weighted average of $\vartheta_j(x_i)$ for node $i$

$$
\langle \vartheta_j(x_i) \rangle_{HMF} = \frac{\frac{1}{N}\sum_{j=1}^N \eta_j^{out} \vartheta_j(x_i)}{\frac{1}{N}\sum_{j=1}^N \eta_j^{out}},
\tag{32}
$$

where $\Gamma^{out} = \left\{ \eta_j^{out} \big| \forall j \in [1, N], \eta_j^{out} = \sum_{i=1}^N \tilde{B}_{(ij)} \right\}$ is the weighted out-degree of node $j$. As reported in Kundu et al. (2022), we can derive all the nodes have statistically identical perturbation affection if the degree correlation is small. Thus, the vector of adversarial perturbation can be approximated as

$$
\Psi(\vartheta) = \frac{\mathbf{1}^T \tilde{B}\vartheta}{\mathbf{1}^T \tilde{B}\mathbf{1}} = \frac{\frac{1}{N}\sum_{j=1}^N \eta_j^{out}\vartheta_j}{\frac{1}{N}\sum_{j=1}^N \eta_j^{out}} = \frac{\langle \Gamma^{out}\vartheta \rangle}{\langle \Gamma^{out} \rangle}.
\tag{33}
$$

From the angle of graph structure, we can presume the variance of components of node set (vectorized as $\vec{x}$) is small. Using HMF approximation theory, we rewrite Equation 5 as

$$
\frac{d\vec{x}}{dt} = \tilde{A}_{(\cdot)}\chi\left(\vec{x}\right) + \overrightarrow{\Gamma^{in}} \odot \varphi\left(\vec{x}, \Psi\left(\vec{x}\right)\right),
\tag{34}
$$

where the sign $\odot$ denotes Hadamard product, i.e. multiply two vectors term by term. There are $(N+1)$ variables with the join of $\Psi\left(\vec{x}\right)$,

$$
\begin{aligned}
\frac{d\Psi\left(\vec{x}\right)}{dt} &= \Psi\left(\tilde{A}_{(\cdot)}\chi\left(\vec{x}\right) + \vec{\Gamma}^{in} \odot \varphi\left(\vec{x}, \Psi\left(\vec{x}\right)\right)\right) \\
&\approx \tilde{A}_{(\cdot)}\chi\left(\Psi\left(\vec{x}\right)\right) + \Psi\left(\vec{\Gamma}^{in}\right)\varphi\left(\Psi\left(\vec{x}\right), \Psi\left(\vec{x}\right)\right)
\end{aligned}
\tag{35}
$$

Thus, let $\tilde{x} = \Psi\left(\vec{x}\right)$, $\tilde{\beta} = \Psi\left(\vec{\Gamma}^{in}\right)$, we can infer the one-dimensional condensed Equation 6.

## F    PROOF OF THEOREM 3.2

Deriving from Theorem 2.6, we know that $\phi\left(\tilde{x}\right) = \frac{H\tilde{x}^2}{\theta\tilde{x}^2+1}$, and $\tilde{x} = \Psi\left(\vec{x}\right)$ is positive, thus $\phi\left(\tilde{x}\right)^T > 0$, $\tilde{x} - M\phi\left(\tilde{x}\right) \geq 0$. Furtherly, $\frac{\phi(\tilde{x})}{\tilde{x}} = \frac{H\tilde{x}}{\theta\tilde{x}^2+1} \leq k$, therefore, let the first-order derivative equal to zero and compute the maximum value, we have $\frac{H\tilde{x}}{\theta\tilde{x}^2+1} \leq H \cdot \left(2\sqrt{\theta}\right)^{-1}$, thus $k \geq H \cdot \left(2\sqrt{\theta}\right)^{-1}$.

## G    PROOF OF THEOREM 3.3

According to Theorem 2.6, we have $\breve{G}(s) = \frac{1}{k} + (1 + s\gamma)G(s) = \frac{1}{k} + (1 - \gamma a)(s + a)^{-1}\tilde{\beta} + \gamma\tilde{\beta}$. First, the $\det\left[\breve{G}(s) + \breve{G}^T(-s)\right]$ is obviously not identically zero. Since $a$ is positive, then $\breve{G}(s)$ is Hurwitz, that is, the poles of all elements of $\breve{G}(s)$ have negative real parts. Pursuant to the claim [Khalil (2002), Theorem 6.1], we calculate

$$
\breve{G}(j\omega) + \breve{G}^T(-j\omega) = \frac{2}{k} + \frac{2\tilde{\beta}\left(a + \gamma\omega^2\right)}{a^2 + \omega^2}.
\tag{36}
$$

Thereby, it is positive definite for all $\omega \in \mathbb{R}$, and $\breve{G}(s)$ is strictly positive real, which further implies Equation 8 has asymptotically-stable equilibrium point.

## H    PERTURBED-BOUND QUANTIFYING

In theory, we can quantify the perturbation bound through the following two steps:

**Step I:** Theorem 3.2 in fact gives the perturbation bounds, wherein the constant $k$, required to be in the interval $\left[H \cdot \left(2\sqrt{\theta}\right)^{-1}, \infty\right)$, reflects the perturbation bounds (Theorem 2.6) when ASEP exists. As we know, for each dataset the parameter $H$ is the average in-degree weight, and directly subject to the perturbation budgets $\rho$, thus we can assume the mapping function as: $H = P(\rho)$. Given the concrete dataset, we have the value $\theta$ according to Table 6. Therefore, we can further infer the detailed interval $\left[P(\rho) \cdot \left(2\sqrt{\theta}\right)^{-1}, \infty\right)$ that constrains the perturbation bounds.

**Step II:** According to Theorem 2.6, for one-dimensional function, the expression $\frac{1}{k} + (1 + s\gamma)G(s)$ is required to be strictly positive, and assume $G(s) = G(\varpi j)$, we can infer $\frac{1}{k} + \mathrm{Re}\left[G(\varpi j)\right] - \gamma\varpi\mathrm{Im}\left[G(\varpi j)\right] > 0$, $\varpi \in [0, \infty]$, where $G(\varpi j) = \mathrm{Re}\left[G(\varpi j)\right] + j\mathrm{Im}\left[G(\varpi j)\right]$. If we regard $\mathrm{Re}\left[G(\varpi j)\right]$ and $\varpi\mathrm{Im}\left[G(\varpi j)\right]$ as two coordinates $x$ and $y$ with $\varpi$ as a parameter, the intermediate variation function (a.k.a. bounded perturbation) must be below the line $\frac{1}{k} + x - \gamma y = 0$, intercepting the point $-\frac{1}{k} + 0j$ in complex plane.

Therefore, from the two steps above, we can quantify the bounds in relation to perturbation budget.

# I IMPLEMENTATION ALGORITHMS

---

**Algorithm 1:** Graph Parameters Calculation

---

**Input:** Adjacency matrix $\widehat{A}$

**Output:** Graph state parameters $\tilde{\beta}$, $\tilde{x}$

**1 Procedure calculate_parameters ($\widehat{A}$)**

2    inDeg ← sum($\widehat{A}$, axis=0)

3    outDeg ← sum($\widehat{A}$, axis=1)

4    degCentrality ← (inDeg + outDeg) / (len($\widehat{A}$)-1)

5    $\tilde{\beta}$ ← sum(outDeg × inDeg) / sum(outDeg)

6    $\tilde{x}$ ← sum(outDeg × degCentrality) /sum(outDeg)

**7 return $\tilde{\beta}$, $\tilde{x}$**

---

---

**Algorithm 2:** Adjacency Matrix Modification

---

**Input:** Adjacency matrix $\widehat{A}$, rate of adversarial perturbation $\widehat{RoP}$

**Output:** Resilience_matrix $\widehat{RA}$

**1 Procedure defense ()**

2    simi_list ← calculate_edge_simi($\widehat{A}$)

3    $\widehat{RA}$ ← $\widehat{A}$

4    critical_$\tilde{\beta}$ ← 0

5    critical_$\tilde{x}$ ← 0

6    Initial_lenth ← len(simi_list)

**7 while** *True* **do**

8      modified_matrix, simi_list ← remove_low_simi_edges($\widehat{RA}$.copy(), simi_list)

9      $\tilde{\beta}$, $\tilde{x}$ ← calculate_parameters(modified_matrix)

10      **if** $\tilde{\beta} > critical\_\tilde{\beta}$ *and* $\tilde{x} > critical\_\tilde{x}$ **then**

11        critical_$\tilde{\beta}$ ← $\tilde{\beta}$

12        critical_$\tilde{x}$ ← $\tilde{x}$

13        $\widehat{RA}$ ← modified_matrix

14      **if** *len(simi_list) ≤ Initial_lenth × $\widehat{RoP}$* **then**

15        break

**16 return $\widehat{RA}$**

---

The inferred asymptotically-stable equilibrium point (ASEP) servers as the referral to purifying the perturbed graph by enforcing adjacency matrix-projection coordinator to be close to such equilibrium point. The implementation mainly involves two phases: i) graph parameters calculation (Algorithm 1). This part aims to compute the two important parameters $\tilde{\beta}$ and $\tilde{x}$ as the referral to modify the adjacency matrix of perturbed graph; and ii) adjacency matrix modification (Algorithm 2). Referring to the two calculated critical-state parameters, this part mainly achieves the modification on adversarial edges. Specifically, we first define function *calculate_edge_simi()* to empirically search the neighboring nodes within $h$-hop ($h$=2), then use Jaccard metric to calibrate the pairwise similarity for each two nodes. Based on the similarity, we define function *remove_low_simi_edges()* to remove those edges with low similarity during the "while" loop, until the inferred coordinator point from the iteratively-modified adjacency matrix approaches to the "Original" point (i.e. ASEP).

## J  EXPERIMENT & PERFORMANCE

### J.1  CONFIGURATION

**Datasets.** Our experiments are performed on five commonly-used scalable realistic graph datasets: Polblogs, Cora, Cora_ML, Citeseer, and Amazon Photo. The statistics are sketched in Table 5. Given our method aims to promote the adversarial resilience from the perspective of graph topology, thus, the dataset Polblogs is completely adaptive since the node feature is unavailable. To sufficiently verify the effectiveness of our proposed method, we also introduce another four feature-involved datasets, however, our method and baselines only concentrate the graph topology information but disregard the node feature in the experiments. Concretely, we set the discrepancy of pairwise-node features as the same in our experimental procedure.

Table 5: Dataset statistics

| Dataset | # Node | # Edge | # Feature | # Classification |
|---------|--------|--------|-----------|------------------|
| PolBlogs | 1222 | 16714 | / | 2 |
| Cora_ML | 2810 | 7981 | 2879 | 7 |
| Cora | 2485 | 5069 | 1433 | 7 |
| Citeseer | 2110 | 3668 | 3703 | 6 |
| Amazon Photo | 7650 | 238162 | 745 | 8 |

**Baselines.** Methodologically, our work aims to improve the graph's adversarial resilience through modifying its adjacency matrix, same as ours, GCN-SVD and GCN also focus on the adjacency-matrix optimization as well. On the other hand, for other non-adjacency-matrix based defense approaches, such as GAT, RGCN, HANG, Mid-GCN, and GOOD-AT, our approach can also boost them from the lens of graph topology, that is, our work can significantly enhance the adversarial resilience by constructing a proper graph structure in advance.

- **GCN (Kipf & Welling, 2017):** Graph Convolutional Network (GCN), as a basic and prevalent deep learning model, is good at handling network/graph-specified data.

- **GAT (Velickovic et al., 2018):** Graph Attention Network (GAT) equipped with attention layers aims to learn differential weights to different neighbors for individuals. It is usually utilized to defend against various graph adversarial attacks.

- **RGCN (Zhu et al., 2019):** It first employs Gaussian distributions as hidden representations of nodes to absorb adversarial changes in the variances of the Gaussian distribution, then assigns different weights to node neighborhoods pursuant to their variances during the convolutions.

- **GCN-SVD (Entezari et al., 2020):** It is a preprocessing method to withstand adversarial attacks by decomposing adjacency matrix of the perturbed graph and then yielding a low-rank approximate matrix to purify the adversarial perturbations (edges).

- **HANG (Zhao et al., 2023):** It is a recently-proposed graph neural flow method, and resorting to the ordinary differential equation to learn a robust representation embeddings for graph nodes.

- **Mid-GCN (Huang et al., 2023):** It is a novel method recently proposed to leverage the mid-frequency signals (Laplacian eigenvalue around 1) on graphs for the promotion of robustness through designing a mid-pass filtering GCN model.

- **GOOD-AT (Li et al., 2024):** Upon the incorporation of OOD detection, the authors leverage popular adversarial training paradigm to defend against poisoning and evasion attacks.

**Graph Adversarial Attacks.** As we know, the distinction between evasion attack and poisoning attack lies in different phases, i.e. the former works in inference process by modifying the testing samples, while the latter works in training phase by modifying the training samples. Thus, the following non-targeted and targeted GAAs used in this paper belong to poisoning attacks:

- **Metattack (Zügner & Günnemann, 2019):** It utilizes meta-learning to generate adversarial perturbations by computing gradients to determine how to modify the graph structure

in order to achieve the attack objective. The gradient computation is typically based on the loss function of the target nodes, aiming to identify which edges should be modified to maximally affect the GNN's prediction results. Based on the computed gradient information, the attacker makes small changes to the graph, such as adding new edges or deleting existing ones. The modified graph structure is then used to retrain the GNN model and evaluate the success of the attack.

- **CE-PGD (Xu et al., 2019):** It uses the negative cross-entropy loss function to measure the difference between the model's predictions and the true labels. The attack starts from the original graph by randomly adding or deleting edges to initialize the adversarial graph. In each iteration, it computes the gradient of the loss function with respect to the current adversarial graph, then updates the graph using the gradient descent method. The updated graph structure is then projected back into the feasible region defined by the attack constraints.

- **DICE (Waniek et al., 2016):** It is a heuristic adversarial attack inspired by modularity, which is used to measure the quality of any given community structure. In other words, it promotes structures where communities have dense connections within themselves and sparse connections between different communities. The algorithm mainly consists of two steps. The first step reduces the connection density among nodes within the same region. The second step increases the connections between nodes in that region and nodes in other regions, by which the algorithm can ignore that region, meaning it will not identify it as a separate community, but instead assign the nodes within that region to other communities.

- **GOttack (Alom et al., 2025):** Unlike the above three non-targeted attacks, it is a targeted attack that aims to target a specific node in a targeted, structural, direct poisoning attack to alter its predicted class, from the perspective of topological structure of graphs.

**Execution Settings.** For each dataset, 10% nodes are randomly chosen for training, 10% nodes for verifying and the rest 80% nodes for testing. For GCN, GAT, HANG, and Mid-GCN, the parameters are kept same as the default settings in their original papers. For GCN-SVD, the reduced rank is tuned from {20, 40, 60, 80, 100}. For our EquiliRes, the ratio of edge modification (deletion/addition) over dissimilar noes is tuned from {1%, 5%, 10%, 15%}, here 1% just for the clean graph, i.e. adversarial perturbation rate is 0%. The experimental results are calculated on average for 10 times. Additionally, the running environment is: ubuntu 20.04.2, GPU: GeForce RTX 3090.

## J.2 EQUILIBRIUM-POINT TRAJECTORY

Pursuant to the parameters as listed in Table 6, we draw the trajectories of asymptotically-stable equilibrium-points on datasets Polblogs, Cora_ML, Cora, Citeseer and Amazon Photo as displayed in Fig. 2-6, wherein the horizontal-longitudinal coordinate converts. From the trajectories, besides the analogous analytics and reviews as presented in Section 5.2, we also obtain the following observations: i) although the plot of adversarial resilience can well-match the attack effect under different rates of adversarial perturbation, the graph's adversarial resilience indeed declines as the strength of perturbation enlarges, which makes sense in intuition, you can image an extreme case that if all edges are removed, the nodes would become isolated and cannot be classified without the connections to each other; and ii) different datasets have distinct trajectories of adversarial resilience, stemming from the difference of graph's internal topology. Thereby, we need carefully figure out such an equilibrium-point trajectory for each unique graph.

## J.3 PERFORMANCE ON ATTACK RESILIENCE

Table 7 and Table 8 show the comparative results separately between our approach and another two adjacency-matrix optimization defense methods GCN and GCN-SVD. Table 9 exhibits the results of GAT, RGCN, HANG, Mid-GCN, and GOOD-AT from the angle of combination with our EquiliRes, from which we can gain the analogous analysis and conclusion as stated in Section 5.3.1.

## J.4 ABLATION

Table 10 and Table 11 respectively exhibit the ablation experiments on datasets Cora_ML and Cora under the three GAAs Metattack, CE-PGD, and DICE.

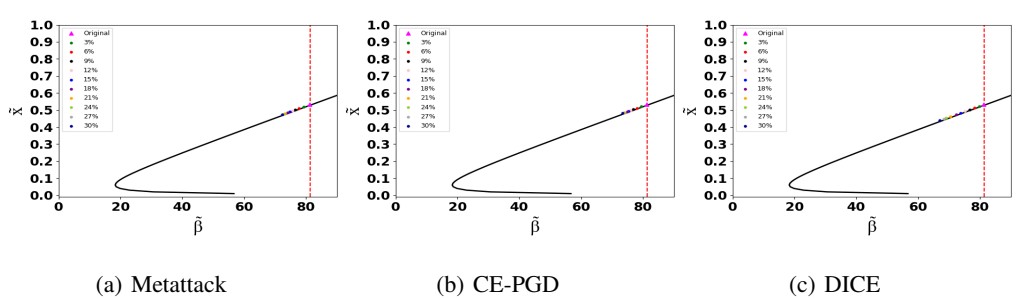

| (a) Metattack | (b) CE-PGD | (c) DICE |

Figure 2: Adversarial effect with Polblogs.

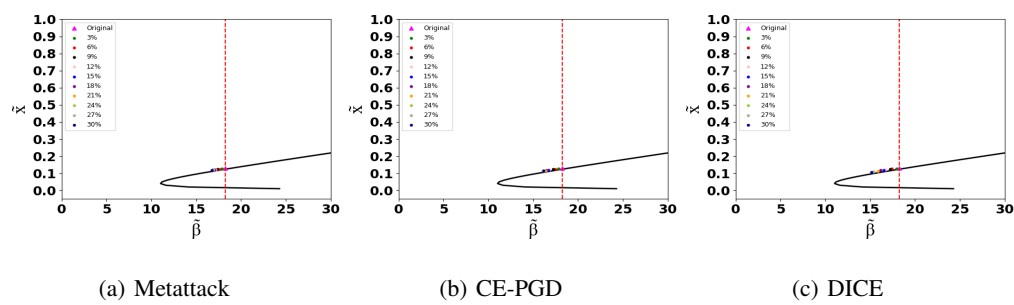

| (a) Metattack | (b) CE-PGD | (c) DICE |

Figure 3: Adversarial effect with Cora_ML.

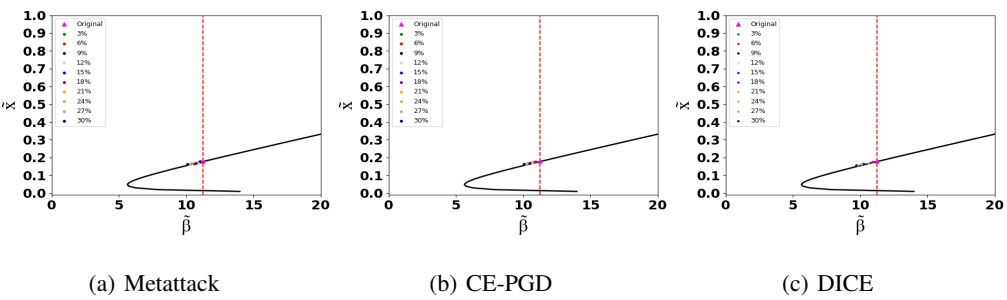

| (a) Metattack | (b) CE-PGD | (c) DICE |

Figure 4: Adversarial effect with Cora.

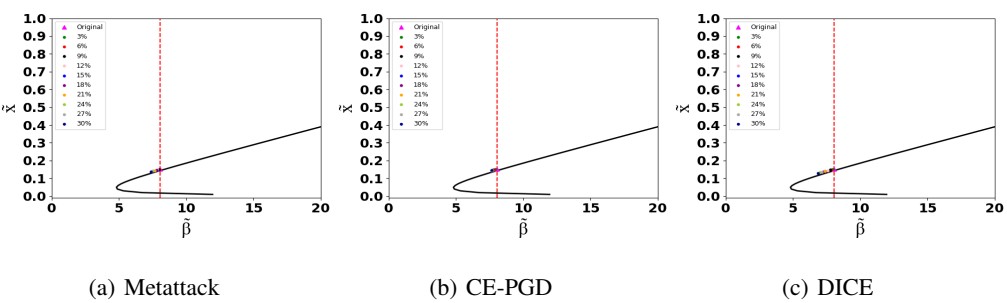

| (a) Metattack | (b) CE-PGD | (c) DICE |

Figure 5: Adversarial effect with Citeseer.

Table 6: Parameters for generating equilibrium-point trajectory

| Dataset | GAA | Parameter |
|---------|-----|-----------|
| Polblogs | Metattack/CE-PGD/DICE | $\theta = 16.6^2$ $\eta$=0.45 |
| Cora_ML | Metattack/CE-PGD/DICE | $\theta = 24^2$ $\eta$=10 |
| Cora | Metattack/CE-PGD/DICE | $\theta = 21^2$ $\eta$=10 |
| Citeseer | Metattack/CE-PGD/DICE | $\theta = 21^2$ $\eta$=10 |
| Amazon Photo | Metattack/CE-PGD/DICE | $\theta = 31.8^2$ $\eta$=8 |

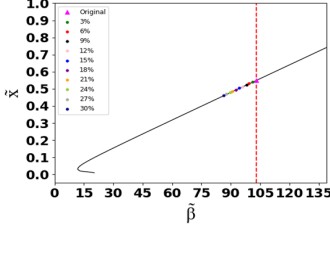 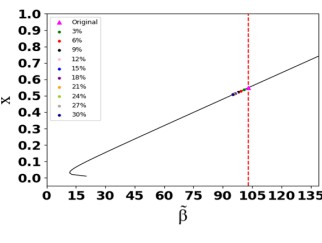 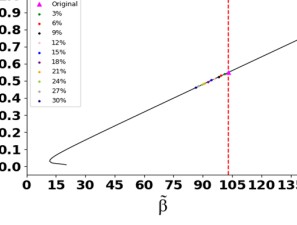

(a) Metattack        (b) CE-PGD        (c) DICE

Figure 6: Adversarial effect with Amazon Photo.

### J.5 RANK OF ADJACENCY MATRIX

The recent research points out the GAAs could enlarge the rank of adjacency matrix quickly, thus, to validate the effectiveness of our EquiliRes, we produce two groups of comparative experiments under the three adversarial attacks, as shown in Fig. 7 on Polblogs and Fig. 8 on Citeseer, to observe the variation of adjacency matrix rank before and after our defense method is engaged.

Seen from the experimental results, we have the following observations: i) as the APR increases, the rank value goes up gradually under the three adversarial attacks on dataset Polblogs, especially with visible lift for Citeseer dataset, this phenomenon shows that graph adversarial attack renders the rank growth to establish connection between dissimilar nodes; ii) correspondingly, our EquiliRes enables to dramatically decline the rank to remove the adversarial edges; and iii) the rank-declining ratio by our proposed EquiliRes has a descending tendency as the perturbation strength becomes larger, this matches the experimental results in Tables 1-3 and Tables 7-9 , reflecting the accuracy of node-classification task would become lower under the gradually growing adversarial perturbations.

Table 7: Accuracy (%) of node classification on Citeseer

| GAA | Per. / Def. | 0% | 5% | 10% | 15% | 20% | 25% |
|-----|-------------|-----|-----|------|------|------|------|
| Metattack | GCN | 54.85±1.71 | 52.30±2.13 | 46.92±1.58 | 43.30±1.42 | 41.18±1.27 | 39.40±0.73 |
| | GCN-SVD | 46.12±1.54 | 45.30±1.64 | 44.89±1.44 | 44.28±1.84 | 43.26±1.45 | 42.84±1.48 |
| | EquiliRes | **55.76±2.07** **(0.91%↑)** | **55.05±1.21** **(2.75%↑)** | **52.59±1.04** **(5.67%↑)** | **50.79±1.78** **(6.51%↑)** | **48.19±0.34** **(4.93%↑)** | **45.15±2.19** **(2.31%↑)** |
| CE-PGD | GCN | 54.85±1.71 | 52.76±1.95 | 50.46±2.44 | 50.34±1.79 | 49.25±1.51 | 49.68±1.46 |
| | GCN-SVD | 46.12±1.54 | 46.22±1.78 | 46.16±1.69 | 45.73±1.93 | 45.80±3.20 | 47.70±3.15 |
| | EquiliRes | **55.76±2.07** **(0.91%↑)** | **54.67±2.44** **(1.91%↑)** | **53.11±2.41** **(2.65%↑)** | **52.87±2.35** **(2.53%↑)** | **51.55±2.32** **(2.30%↑)** | **52.89±2.18** **(3.21%↑)** |
| DICE | GCN | 54.85±1.71 | 51.48±1.48 | 51.16±1.75 | 49.00±1.43 | 47.71±1.48 | 45.94±1.53 |
| | GCN-SVD | 46.12±1.54 | 44.70±1.96 | 41.65±3.07 | 41.58±2.43 | 39.61±2.84 | 37.71±1.75 |
| | EquiliRes | **55.76±2.07** **(0.91%↑)** | **53.30±2.09** **(1.82%↑)** | **51.42±1.73** **(0.26%↑)** | **50.81±2.16** **(1.81%↑)** | **49.46±1.78** **(1.75%↑)** | **47.01±1.68** **(1.07%↑)** |

Table 8: Accuracy (%) of node classification on Amazon Photo

| GAA | Per.
Def. | 0% | 5% | 10% | 15% | 20% | 25% |
|---|---|---|---|---|---|---|---|
| Meta
ttack | GCN | 92.15±0.34 | 82.42±7.74 | 82.81±5.81 | 79.85±13.16 | 62.95±14.96 | 61.08±15.26 |
| | GCN-SVD | 87.13±0.47 | 84.32±7.94 | 82.01±9.81 | 72.39±12.46 | 71.21±12.29 | 63.57±16.78 |
| | EquiliRes | **92.87±0.46**
(**0.72%**↑) | **91.65±0.21**
(**7.33%**↑) | **85.11±20.05**
(**2.30%**↑) | **82.25±30.00**
(**2.40%**↑) | **74.35±31.92**
(**3.14%**↑) | **81.14±25.58**
(**17.57%**↑) |
| CE-
PGD | GCN | 92.15±0.34 | 90.42±0.37 | 90.15±0.31 | 90.12±0.29 | 88.41±0.56 | 88.29±0.34 |
| | GCN-SVD | 87.13±0.47 | 85.82±4.83 | 87.75±0.36 | 86.78±2.47 | 86.75±2.17 | 86.17±0.49 |
| | EquiliRes | **92.87±0.46**
(**0.72%**↑) | **91.12±0.32**
(**0.70%**↑) | **90.62±0.45**
(**0.47%**↑) | **90.33±0.17**
(**0.21%**↑) | **89.39±0.46**
(**0.98%**↑) | **88.58±0.25**
(**0.29%**↑) |
| DICE | GCN | 92.15±0.34 | 90.96±0.65 | 89.03±5.30 | 89.00±1.01 | 86.60±3.17 | 84.90±3.18 |
| | GCN-SVD | 87.13±0.47 | 85.03±0.56 | 82.74±0.89 | 81.54±0.41 | 75.75±2.72 | 76.14±0.57 |
| | EquiliRes | **92.87±0.46**
(**0.72%**↑) | **91.09±0.33**
(**0.13%**↑) | **91.45±0.21**
(**2.42%**↑) | **90.84±0.12**
(**1.84%**↑) | **89.45±0.53**
(**2.85%**↑) | **88.59±0.11**
(**3.69%**↑) |

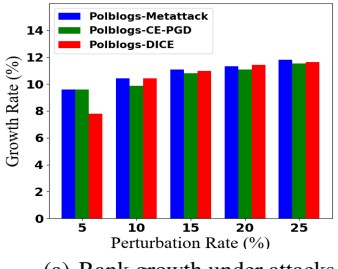
(a) Rank growth under attacks

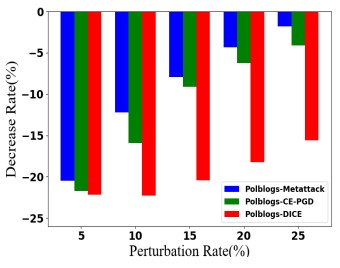
(b) Rank decrease by EquiliRes

Figure 7: adjacency matrix of Polblogs.

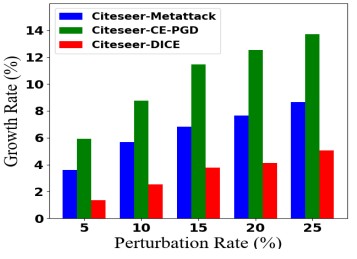
(a) Rank growth under attacks

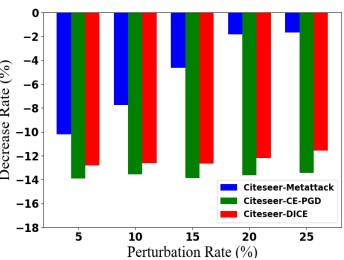
(b) Rank decrease by EquiliRes

Figure 8: adjacency matrix of Citeseer.

Table 9: Accuracy (%) of node classification under combination with EquiliRes using Metattack

| Dataset | Per.
Def. | 0% | 5% | 10% | 15% | 20% | 25% |
|---|---|---|---|---|---|---|---|
| Cora
_ML | GAT | 77.55±1.15 | 68.86±1.04 | 64.13±1.37 | 60.65±0.92 | 58.75±1.70 | 56.43±1.26 |
| | Ours+GAT | **77.69±1.30**
**(0.14%↑)** | **71.15±1.85**
**(2.29%↑)** | **68.32±1.99**
**(4.19%↑)** | **66.11±1.61**
**(5.46%↑)** | **64.52±1.71**
**(5.77%↑)** | **62.12±1.59**
**(5.69%↑)** |
| | RGCN | 80.57±0.18 | 70.43±0.32 | 62.38±0.29 | 55.78±0.34 | 52.14±0.24 | 48.10±0.28 |
| | Ours+RGCN | **80.98±0.14**
**(0.41%↑)** | **71.48±0.27**
**(1.05%↑)** | **66.04±0.32**
**(3.66%↑)** | **59.87±0.23**
**(4.09%↑)** | **57.07±0.19**
**(4.93%↑)** | **52.62±0.23**
**(4.52%↑)** |
| | HANG | 64.11±4.49 | 57.98±2.30 | 52.98±2.23 | 50.54±2.52 | 47.64±2.51 | 45.93±2.93 |
| | Ours+HANG | **64.34±5.03**
**(0.23%↑)** | **60.47±3.58**
**(2.49%↑)** | **57.08±2.68**
**(4.10%↑)** | **55.34±3.14**
**(4.80%↑)** | **54.23±3.63**
**(6.59%↑)** | **53.06±3.25**
**(7.13%↑)** |
| | Mid-GCN | 83.40±0.21 | 80.01±0.37 | 78.95±0.37 | 77.80±0.55 | 75.66±0.57 | 74.17±0.72 |
| | Ours+Mid-
GCN | **83.49±0.16**
**(0.09%↑)** | **80.22±0.46**
**(0.21%↑)** | **79.70±0.34**
**(0.75%↑)** | **78.03±0.63**
**(0.23%↑)** | **76.19±0.72**
**(0.53%↑)** | **74.53±0.83**
**(0.36%↑)** |
| Cora | GAT | 73.28±1.43 | 67.53±1.39 | 61.20±1.77 | 58.09±1.21 | 55.07±1.59 | 52.16±1.75 |
| | Ours+GAT | **73.43±1.36**
**(0.15%↑)** | **67.65 ±1.60**
**(0.12%↑)** | **63.53±2.06**
**(2.33%↑)** | **60.08±2.29**
**(1.99%↑)** | **57.66±2.22**
**(2.59%↑)** | **55.23±2.05**
**(3.07%↑)** |
| | RGCN | 77.09±0.29 | 72.57±0.26 | 65.75±0.28 | 57.26±0.56 | 55.67±0.35 | 49.33±0.46 |
| | Ours+RGCN | **77.45±0.37**
**(0.36%↑)** | **73.35±0.18**
**(0.78%↑)** | **65.87±0.47**
**(0.12%↑)** | **58.57±0.49**
**(1.31%↑)** | **56.03±0.43**
**(0.36%↑)** | **49.95±0.38**
**(0.62%↑)** |
| | HANG | 58.22±4.79 | 55.23±3.80 | 50.25±3.30 | 48.36±3.33 | 45.84±3.12 | 44.54±3.14 |
| | Ours+HANG | **58.40±4.82**
**(0.18%↑)** | **55.49±3.98**
**(0.26%↑)** | **52.41±3.50**
**(2.16%↑)** | **50.22±3.58**
**(1.86%↑)** | **48.77±2.99**
**(2.93%↑)** | **47.29±2.85**
**(2.75%↑)** |
| | Mid-GCN | 83.10±0.41 | 75.81±15.82 | 78.88±0.93 | 75.43±1.11 | 75.50±1.35 | 74.43±0.83 |
| | Ours+Mid-
GCN | **83.23±0.36**
**(0.13%↑)** | **81.27±0.72**
**(5.46%↑)** | **80.20±1.10**
**(1.32%↑)** | **77.69±0.82**
**(2.26%↑)** | **76.74±0.64**
**(1.24%↑)** | **75.16±0.76**
**(0.73%↑)** |
| | GOOD-AT | 84.12±0.34 | 82.73±0.78 | 81.31±1.65 | 80.20±1.24 | 79.15±0.36 | 78.88±0.85 |
| | Ours+GOOD-
AT | **84.30±0.56**
**(0.18%↑)** | **83.21±0.63**
**(0.48%↑)** | **81.99±0.65**
**(0.68%↑)** | **81.32±0.41**
**(1.12%↑)** | **80.30±0.43**
**(1.15%↑)** | **80.46±0.34**
**(1.58%↑)** |
| Citeseer | GAT | 64.04±1.85 | 60.21±1.91 | 54.00±1.81 | 49.92±2.02 | 46.96±1.70 | 44.65±0.97 |
| | Ours+GAT | **64.17±1.73**
**(0.13%↑)** | **61.14±1.97**
**(0.93%↑)** | **54.60±1.65**
**(0.60%↑)** | **53.21±1.36**
**(3.29%↑)** | **51.30±1.64**
**(4.34%↑)** | **48.39±2.31**
**(3.74%↑)** |
| | RGCN | 67.53±0.49 | 63.49±0.35 | 57.22±0.39 | 51.91±0.33 | 50.37±0.65 | 45.34±0.70 |
| | Ours+RGCN | **67.72±0.58**
**(0.19%↑)** | **63.50±0.43**
**(0.01%↑)** | **58.10±0.32**
**(0.88%↑)** | **51.98±0.34**
**(0.07%↑)** | **50.49±0.53**
**(0.12%↑)** | **45.63±0.47**
**(0.29%↑)** |
| | HANG | 56.48±2.85 | 52.92±1.91 | 48.05±2.72 | 45.12±0.83 | 44.29±1.83 | 40.61±1.96 |
| | Ours+HANG | **56.54±2.74**
**(0.06%↑)** | **53.17±2.37**
**(0.25%↑)** | **49.62±2.17**
**(1.57%↑)** | **49.62±2.17**
**(4.50%↑)** | **46.64±1.60**
**(2.35%↑)** | **44.70±2.40**
**(4.09%↑)** |
| | Mid-GCN | 74.52±0.32 | 72.37±0.67 | 71.05±0.66 | 65.84±1.06 | 67.00±0.52 | 63.06±0.99 |
| | Ours+Mid-
GCN | **74.66±0.47**
**(0.14%↑)** | **72.70±0.37**
**(0.33%↑)** | **71.52±0.71**
**(0.47%↑)** | **66.45±1.03**
**(0.61%↑)** | **67.84±0.50**
**(0.84%↑)** | **63.84±1.25**
**(0.78%↑)** |
| | GOOD-AT | 74.33±0.23 | 71.71±0.55 | 70.17±1.28 | 68.88±0.84 | 68.22±1.60 | 66.58±2.68 |
| | Ours+GOOD-
AT | **74.51±0.27**
**(0.18%↑)** | **72.41±0.27**
**(0.70%↑)** | **70.62±2.07**
**(0.45%↑)** | **70.50±1.31**
**(1.62%↑)** | **70.03±1.72**
**(1.81%↑)** | **68.12±2.75**
**(1.54%↑)** |
| Amazon
Photo | GAT | 93.54±0.18 | 91.73±0.39 | 76.53±31.14 | 69.99±32.95 | 51.57±38.85 | 62.53±33.59 |
| | Ours+GAT | **93.70±0.16**
**(0.16%↑)** | **91.74±0.18**
**(0.01%↑)** | **83.65±20.01**
**(7.12%↑)** | **76.19±29.97**
**(6.20%↑)** | **74.30±31.89**
**(22.73%↑)** | **81.05±25.55**
**(18.52%↑)** |
| | HANG | 93.47±0.32 | 91.62±0.54 | 91.51±0.39 | 90.93±0.68 | 90.16±0.56 | 89.92±0.61 |
| | Ours+HANG | **93.63±0.43**
**(0.16%↑)** | **92.64±0.44**
**(1.02%↑)** | **91.72±0.35**
**(0.21%↑)** | **91.97±0.42**
**(1.04%↑)** | **91.04±0.39**
**(0.88%↑)** | **90.85±0.73**
**(0.93%↑)** |
| | Mid-GCN | 77.25±0.48 | 74.39±0.49 | 65.95±0.82 | 57.08±1.70 | 48.75±0.88 | 46.75±1.13 |
| | Ours+Mid-
GCN | **78.45±0.50**
**(1.20%↑)** | **74.81±0.82**
**(0.42%↑)** | **66.15±1.14**
**(0.20%↑)** | **59.47±1.18**
**(2.39%↑)** | **51.02±1.35**
**(2.27%↑)** | **47.37±1.43**
**(0.62%↑)** |

Table 10: Accuracy (%) of node classification with/without referring to ASEP on Cora_ML

| GAA | Per. | 0% | 5% | 10% | 15% | 20% | 25% |
|---|---|---|---|---|---|---|---|
| Metattack | w/o | 74.31±0.50 | 67.15±1.00 | 59.42±0.66 | 53.83±0.97 | 49.98±0.66 | 46.49±0.92 |
| | w/ | **74.77±1.49** (**0.46%↑**) | **69.12±1.12** (**1.97%↑**) | **65.62±0.96** (**6.20%↑**) | **62.70±1.32** (**8.87%↑**) | **60.72±0.61** (**10.74%↑**) | **58.39±1.12** (**11.90%↑**) |
| CE-PGD | w/o | 74.31±0.50 | 70.81±0.36 | 69.73±0.33 | 69.72±0.30 | 68.76±0.27 | 68.28±0.32 |
| | w/ | **74.77±1.49** (**0.46%↑**) | **72.77±1.37** (**1.96%↑**) | **72.65±1.26** (**2.92%↑**) | **72.65±2.27** (**2.93%↑**) | **71.98±2.80** (**3.22%↑**) | **71.70±2.30** (**3.42%↑**) |
| DICE | w/o | 74.31±0.50 | 70.63±0.38 | 48.77±0.28 | 66.83±0.65 | 64.34±0.41 | 61.17±0.33 |
| | w/ | **74.77±1.49** (**0.46%↑**) | **72.43±1.83** (**1.80%↑**) | **69.53±1.38** (**20.76%↑**) | **67.52±1.38** (**0.69%↑**) | **65.74±1.30** (**1.40%↑**) | **63.58±1.58** (**2.41%↑**) |

Table 11: Accuracy (%) of node classification with/without referring to ASEP on Cora

| GAA | Per. | 0% | 5% | 10% | 15% | 20% | 25% |
|---|---|---|---|---|---|---|---|
| Metattack | w/o | 62.05±0.83 | 61.30±0.56 | 56.44±1.26 | 51.44±0.54 | 51.04±0.39 | 46.01±0.55 |
| | w/ | **67.74±2.65** (**5.69%↑**) | **64.26±0.61** (**2.96%↑**) | **59.02±1.28** (**2.58%↑**) | **57.31±2.05** (**5.87%↑**) | **54.78±0.77** (**3.74%↑**) | **52.04±0.82** (**6.03%↑**) |
| CE-PGD | w/o | 62.05±0.83 | 65.15±1.21 | 64.37±0.63 | 62.31±1.18 | 61.07±0.83 | 60.61±0.92 |
| | w/ | **67.74±2.65** (**5.69%↑**) | **67.55±3.69** (**2.40%↑**) | **65.64±3.81** (**1.27%↑**) | **63.29±3.11** (**0.98%↑**) | **62.81±3.24** (**1.74%↑**) | **62.27±3.69** (**1.66%↑**) |
| DICE | w/o | 62.05±0.83 | 59.35±1.01 | 58.07±0.71 | 54.61±0.56 | 53.14±1.05 | 51.60±0.86 |
| | w/ | **67.74±2.65** (**5.69%↑**) | **66.67±2.80** (**7.32%↑**) | **63.58±3.13** (**5.51%↑**) | **61.77±2.95** (**7.16%↑**) | **60.41±3.17** (**7.27%↑**) | **57.78±2.63** (**6.18%↑**) |

## J.6 SINGULAR-VALUE VARIATION

Jin et al. (2020) disclose the attack Metattack can enlarge the singular values of adjacency matrix, thus we run a set of experiments to showcase the variation of singular values of adjacency matrix after using our EquiliRes on moderate-scale Polblogs and large-scale Amazon Photo datasets, and depict the results in Fig. 9 and Fig. 10, from which the following observations can be obtained.

First, our EquiliRes indeed reduces the singular values of adjacency matrices under the three adversarial attacks on both datasets, which indicates our proposed equilibrium-point based modification fashion is reasonable and valid to remove adversarial edges w.r.t. different-level adversarial perturbations. In other words, the process of modifying the adjacency matrix to approach to the equilibrium state exactly corresponds to the manipulation of declining singular values. Second, different types of graph adversarial attacks with different attack strengths cause the singular values to have different-level lifts. Correspondingly, our EquiliRes also results in different-level descent in parallel.

## J.7 COMPUTATION OVERHEAD AND SCALABILITY

One advantages of our EquiliRes lies in post-learning that can be obtained from Laplace Transfer without executing deep-learning manipulation in reality, i.e., the critical state of adversarial resilience of graph regime, stemming from the theoretic guarantee of asymptotic stability. Furthermore, our method employs the condensed one-dimensional function to sketch the entire graph' dynamics via degree centrality, rather than other sophisticated multi-dimensional functions, which also decreases the running time to a large extent. To comparatively analyze the efficiency, we calculate the time overhead from the training input to defense output with the experiment settings: APR is set from 0% to 25%, the designated running epoch is set as 200 rounds, and the time overhead is counted on average for 5 times for each experiment. The results on the four datasets Polblogs, Cora_ML, Cora, and Citeseer under the attacks CE-PGD are plotted in Fig. 11.

From the experimental results, we can observe that: i) compared to GAT, HANG, Ours+GAT, and Ours+HANG, the basic GCN, GCN-SVD and our EquiliRes take less time cost on the datasets, this is because GCN only needs to capture the graph structure information without other handling. The singular value decomposition makes GCN-SVD cost similar or a little more overhead than GCN, this is because the decomposition only causes the graph to remove some edges. For our

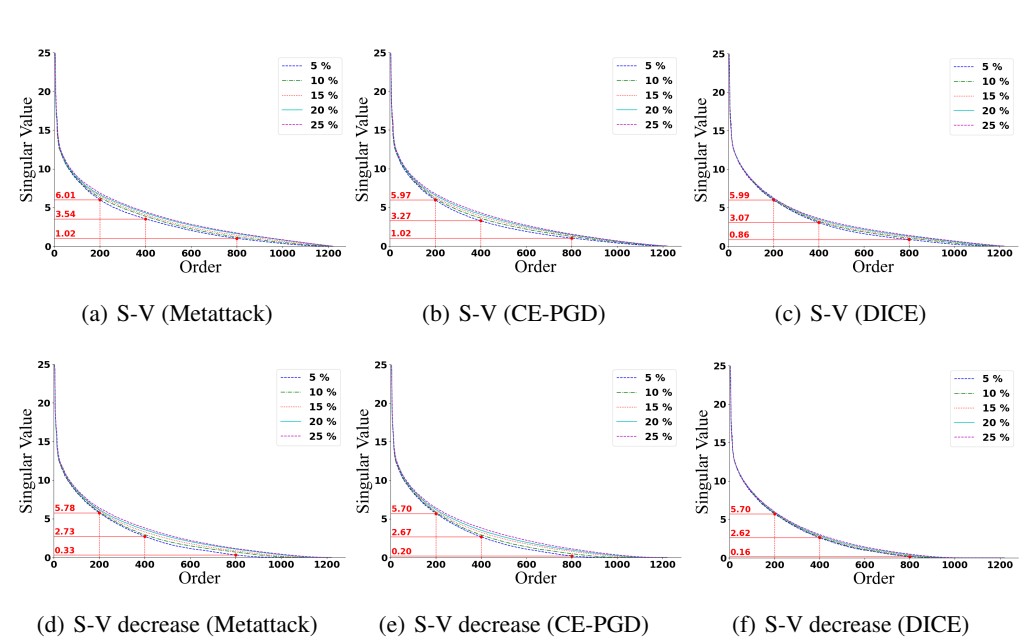

Figure 9: Singular-Value (S-V) variation of Polblogs with EquiliRes.

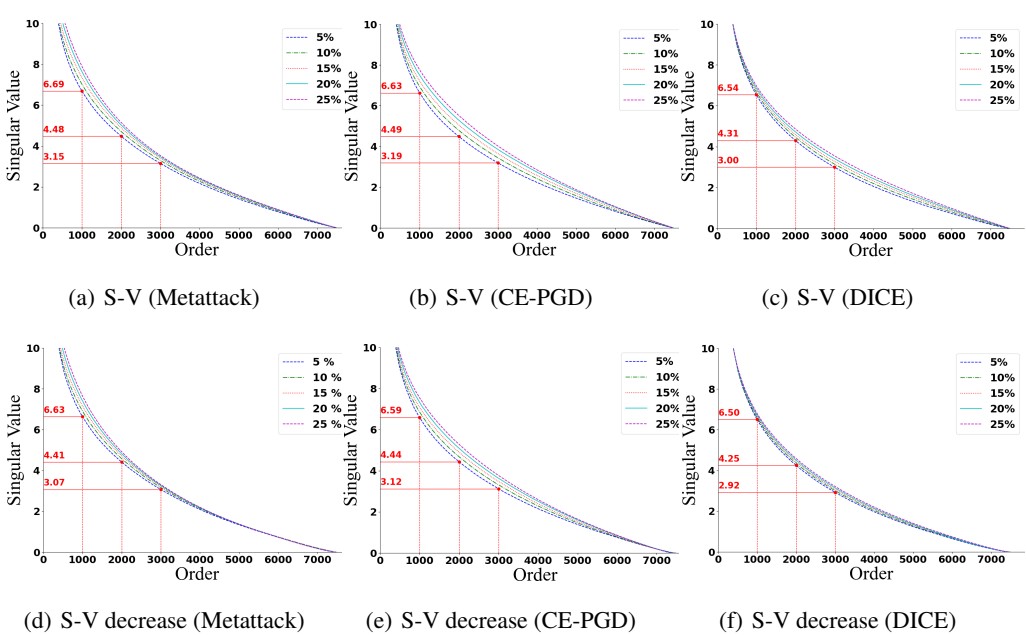

Figure 10: Singular-Value (S-V) variation of Amazon Photo with EquiliRes.

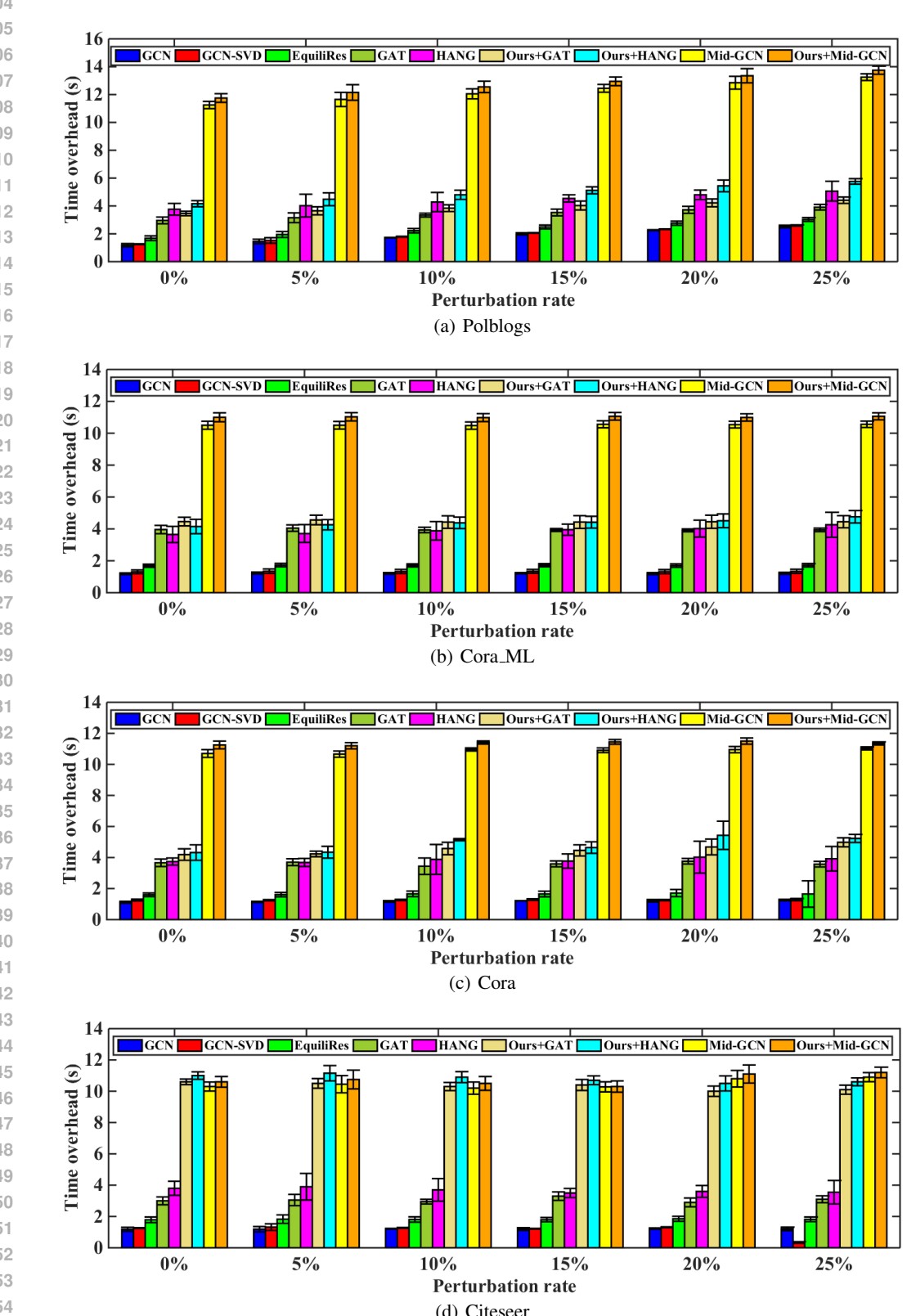

Figure 11: Computation overhead under CE-PGD.

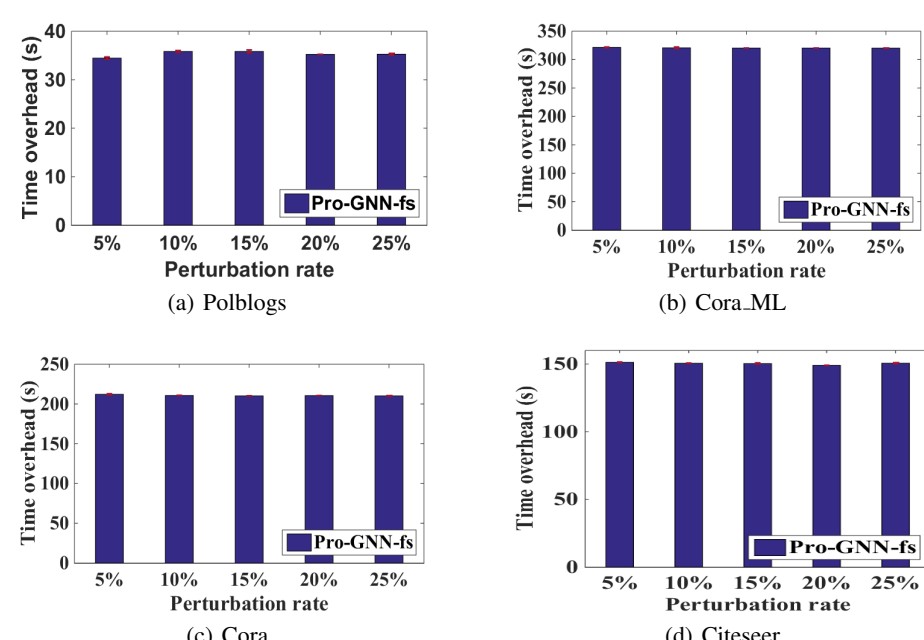

Figure 12: Time overhead of Pro-GNN-fs under CE-PGD.

EquiliRes, due to the extra location on equilibrium points, thus it costs a little more than GCN-SVD; ii) Compared with other methods, Mid-GCN and Ours+Mid-GCN consume more time remarkably, we think this is because they need capture the mid-frequency signals from the high-frequency and low-frequency signals through leveraging a filter for graphs, which costs relatively-long time; and iii) although Ours+GAT, Ours+HANG, and Ours+Mid-GCN need a little more overhead than respective individuals, nevertheless, the introduction of our method would not bring too much cost because of the pre-location of inferred equilibrium-point in theory. Overall, these methods are all lightweight compared to those deep learning-based adjacency-matrix iterative optimization mechanisms, such as Pro-GNN-fs (Jin et al., 2020). As shown in Fig. 12, Pro-GNN-fs costs much more time than the others, deriving from the repeatedly search for the optimal low-rank adjacency matrix. For instance, the maximum overhead of Pro-GNN-fs exceeds 300 seconds on dataset Cora_ML, while the maximum of the others is no more than 14 seconds. Comparatively analyzing from the two groups of computation overhead, we know that all these adjacency matrix-based and mid-frequency signal-captured defense approaches cost much less time than the deep learning-based adjacency-matrix iterative optimization methods.

**Scalability of Computational Overhead.** Given the purification operations of these adversarial defense methods are all edged-centered, thus, we calculate the computational overhead as the number of edges enlarges regarding the four scalable datasets, and the experimental results are listed in Table 12, from which we can see that the proposed EquiliRes is lightweight like GCN and GCN-SVD, compared to other baselines like Hang, Mid-GCN.

## K    RELATED WORK

### K.1    GRAPH ADVERSARIAL ATTACK

Currently, extensive research has witnessed the graph data is suffering from severe adversarial attacks (Wang & Gong, 2019; Liu et al., 2019; Bojchevski & Günnemann, 2019), partially deriving from the sophisticated dependency (links) among graph nodes. For instance, from the viewpoint of global graph, the work Wang & Gong (2019) manipulates graph structure towards the collective classification question. Metattack (Zügner & Günnemann, 2019) leverages meta-learning to strategically attack the edges, in addition to modify weight, and the reference Zügner & Günnemann

Table 12: Time overhead (s) at the rate of adversarial perturbation 15% (Ns, Es respectively denote Nodes and Edges)

| Dataset | Citeseer (2110Ns 3668Es) | Cora (2485Ns 5069Es) | Cora_ML (2810Ns 7981Es) | Polblogs (1222Ns 16714Es) |
|---|---|---|---|---|
| GCN | 1.20±0.09 | 1.18±0.01 | 1.22±0.04 | 1.37±0.06 |
| GCN-SVD | 1.22±0.02 | 1.29±0.05 | 1.50±0.11 | 1.42±0.01 |
| EquiliRes | 1.80±0.14 | 1.65±0.18 | 1.74±0.11 | 1.87±0.12 |
| HANG | 3.50±0.29 | 3.77±0.46 | 3.95±0.35 | 4.06±0.24 |
| Ours+HANG | 10.70±0.28 | 4.64±0.37 | 4.43±0.38 | 4.51±0.23 |
| GAT | 3.30±0.27 | 3.59±0.19 | 3.93±0.09 | 3.31±0.22 |
| Ours+GAT | 10.40±0.35 | 4.46±0.36 | 4.46±0.42 | 3.66±0.30 |
| Mid-GCN | 10.28±0.32 | 10.91±0.15 | 10.58±0.23 | 11.75±0.26 |
| Ours+Mid-GCN | 10.30±0.36 | 11.45±0.15 | 11.08±0.25 | 12.25±0.29 |

(2019) reported that Metattack could quickly boost the rank of adjacency matrix, and tended to establish connections between dissimilar nodes. Xu et al. (2019) employ negative cross entropy-based PGD (CE-PGD) and CW attack-based PGD (CW-PGD) to generate topology perturbations. From the angle of network community, DICE (Waniek et al., 2016) launches an attack on social networks through deliberately removing internal connectivity and building external connectivity to degrade the effectiveness of node classification. Recently, GOttack (Alom et al., 2025) aims to target a specific node in a targeted, structural, direct poisoning attack to alter its predicted class, from the perspective of topological structure of graphs.

More seriously, towards the training graph data, even inaccessible, some crucial properties (node degree, subgraph structure) still can be explored by property inference attack (PIA) (Wang & Wang, 2022) and structure membership inference attack (SMIA) (Wang & Wang, 2024). He et al. (2021) propose how to steal graph links from training dataset given a non-access scenario to GNN model. Generally speaking, large graph regime can be usually deemed as a sophisticated system, the complexity of interaction (information-passing) fashion among a large number of nodes makes it difficult to defend against continuous and imperceivable adversarial perturbations, especially for large-scale graph regimes.

### K.2 GRAPH ADVERSARIAL DEFENSE

At present, the adversarial defense is studied mainly from two lines: graph per se and graph neural flow. The former mainly improves the adversarial resilience through modifying topology and node feature, for example, from the preprocessing perspective, Wu et al. (2019) firstly remove the dissimilar edges resorting to node feature-based Jaccard-similarity comparison with a preset threshold. GCN-SVD (Chamberlain et al., 2021b), serving as a preprocessing means as well, utilizes SVD to decompose adjacency matrix of the perturbed graph and acquires a low-rank adjacency-matrix approximation to represent a cleaner graph. Pro-GCN-fs (Jin et al., 2020) tries to learn a low-rank and close adjacency matrix to the original. Obviously, the above methods all aim to modify the graph topology and/or features to promote adversarial resilience. Our work falls into this category due to the consideration of finding out the intrinsic critical state of adversarial resilience through modifying topology. More concretely, our work only focuses on graph structure without considering node feature.

The other line, i.e. neural ODE network, aims to improve the adversarial resilience through studying the graph neural flow (GNF), that is, neural network constrains the input and output to adhere to certain physical laws, and the transmission of such constrained data within the neural network is referred to as GNF. Typically, these constraints are implemented through system equations from physics. In other words, GNNs can be instantiated with system equations to learn graph data. In recent years, the neural ODE has been successfully applied to GNN through modeling information exchange. The studies Chamberlain et al. (2021b); Thorpe et al. (2022) deem message-passing process as the heat diffusion model, while the references Chamberlain et al. (2021a); Song et al. (2022) regard the message-passing process as Beltrami diffusion. In addition, towards graph nodes,

Rusch et al. (2022) adopt the oscillators to model nodes and conduct the message-passing process under the guidance of a coupled oscillating ODE. Recently, HANG (Zhao et al., 2023) imposes an energy-conservation constraint on GNF, endowed with Lyapunov stability, to enhance the adversarial robustness of GNNs. Recently, upon the incorporation of OOD detection, GOOD-AT (Li et al., 2024) leverages the popular adversarial training paradigm to defend against poisoning and evasion attacks. Differently, our approach models graph adversarial perturbation by adhering to the asymptotic stability of Lyapunov theory, simulating the graph state change under adversarial attacks. By observing the phenomena of dynamic variation, we propose a preprocessing graph optimization method tailored for graph data through resorting to equilibrium-point theory in dynamic systems. This category of GNF-based defense methods is designed to construct graph neural flows for GNNs, while our work is oriented toward graph data processing.

## L  MOTIVATION AND OPEN ISSUES

**Motivation.** Towards why we need dynamic-system findings to study adversarial learning, We think there are at least three reasons: i) the oscillation in dynamic system and the perturbation in adversarial learning have common characteristic-dynamics. Considering the difficulty of measuring adversarial attacks (To our best knowledge, no solution to quantify adversarial attacks to date), we can employ Laplace Transform to measure adversarial attack quantitatively, especially for the scenarios where we have no idea about the behavior of GAAs. Hence, resorting to the findings in dynamic system to bridge connection to adversarial learning is proper and worth in-depth exploration, and Fig. 2-6 show the consistency between the equilibrium-point trajectory and adversarial attack effect; ii) using the stability theory in dynamic systems, we can infer an optimal robust (equilibrium-point) state in theory for each graph regime in a simple yet valid manner; and iii) in the light of equilibrium point, we can directly purify the perturbed adjacency matrix in a light-weight way, resulting in a lower time overhead compared to the time-consuming model-retraining methods (e.g. Pro-GNN-fs) as shown in Fig. 11 and Fig. 12.

**Open Issues.** Our work is completely a new attempt to study graph adversarial resilience, and the following issues are well worthy discussing: i) **Condensed One-Dimensional Mapping Function.** In our work, a simple condensed one-dimensional function is developed to map the dynamics of graph regime as a whole, in fact, two-dimensional mapping or multi-dimensional mapping can be explored as well for distinct graph regimes, this is because our proposed asymptotically-stable theoretic framework fully supports multi-dimensional dynamic-variation function. The more accurate the modeling on GAAs, the higher the adversarial resilience would be; ii) **Adversarial Resilience on Multi-Modal Data.** This paper only concentrates graph data, a natural question emerges, i.e. whether other modal data also has such equilibrium state to maintain the adversarial resilience, such as image, audio, text, etc. If exist, whether we can tackle in the same way in technique given the consideration of totally different data structures; and iii) **Parameters of Adversarial Resilience-Declining Function.** As shown in Fig. 2-6, different graph regimes usually have distinct adversarial resilience-declining tendencies under different-level perturbations, hence, adequate parameters are needed to well-fit the variation of adverseness by various graph adversarial attacks. Only the well-fitting between theoretical equilibrium-point trajectory and adversarial-effect dots can truthfully unveil the behavior of adversarial attack.

