# OpenReview forum: "Adverseness vs. Equilibrium: Exploring Graph Adversarial Resilience through Dynamic Equilibrium"
_ICLR.cc/2026/Conference — Submitted to ICLR 2026_

### Official Review · Reviewer_9ezX · 2025-10-25

**Soundness:** 2
**Presentation:** 2
**Contribution:** 2
**Rating:** 4
**Confidence:** 3

**Summary:**

This paper introduces an approach to graph adversarial defense by drawing inspiration from the theory of complex dynamical systems. The authors propose the EquiliRes framework, which uses concepts like Lyapunov stability and Laplace transforms to prove the existence of this equilibrium. The framework models the graph’s dynamics with a condensed one-dimensional function based on degree centrality and iteratively modifies the adjacency matrix to move the graph toward the equilibrium point, thereby “purifying” it from adversarial perturbations.

**Strengths:**

1.The paper’s approach is highly original, combining concepts from dynamical systems and graph adversarial learning in a novel way. This cross-disciplinary perspective could potentially open up new research directions.

2.The paper tackles a bold, high-level idea with the potential to transform the way we think about graph adversarial defense. The theoretical framework, if fully validated, could offer a profound contribution.

**Weaknesses:**

1.The connection between the theoretical framework (Lyapunov stability and Laplace transforms) and the concrete implementation based on degree centrality is not adequately justified. This “leap of faith” leaves a critical gap in understanding why such a simplification is both valid and effective for adversarial defense in graphs.

2.The empirical comparison is insufficient, primarily involving older and simpler baselines like GCN and GCN-SVD. To demonstrate the true effectiveness of the method, comparisons to more recent and robust defense methods would be necessary.

**Questions:**

1.Could the authors provide a clearer, step-by-step explanation of how this simplification is both valid and sufficient to capture the dynamics of graph adversarial perturbations?

2.Why were more recent, state-of-the-art defense methods such as Pro-GNN not included in the comparisons? How does EquiliRes perform in direct comparison to these methods?

3.The method focuses on purifying graph topology. Does the theoretical framework provide any insight into defending against perturbations to node features as well?

---

> ### Author Response · Authors · 2025-11-22
> **Reply to Q1**
>
> **Q1.** Could the authors provide a clearer, step-by-step explanation of how this simplification is both valid and sufficient to capture the dynamics of graph adversarial perturbations?
>
> **Reply:**
> The core of this paper lies in enhancing the graph’s robustness by locating the asymptotically stable equilibrium point (ASEP) from the angle of topology.
>
> Concretely, there are four steps:
>
> **Step I: Bridging Connection between Adversarial Learning and Complex Dynamic System**
>
> Consider the limited studies on adversarial learning in the facets of fundamental theory and interpretability to date, we attempt to bridge the connection between adversarial perturbations and oscillation behavior of complex dynamic system (CDS), then use the Lyapunov Stability Theory in CDS to interpret the adversarial attack/perturbation behavior in machine learning. Upon which, we propose the conjecture that the ASEP in CDS corresponds to the critical state of adversarial resilience.
>
> **Step II: Modeling for Adversarial Perturbations**
>
> Given the conjecture, we need figure out adaptive function (Eq. 1) to describe the concrete adversarial attacks, and accurately quantify the perturbations with Laplace Transform (Eqs. 2-3). Resorting to such modeling function on adversarial attack and Laplace Transform-based intermediate function (Eq. 4), we propose a rigor and complete theoretical framework (Theorem 2.6) for graph data, this theoretical framework provides the prerequisite for the existence of ASEP from the perspective of N-dimensional space.
>
> **Step III: Condensing High-dimensional Data into One-Dimensional Expression**
>
> Consider the expensive computational overhead for high-dimensional graph data, especially today’s large-scale networks/graphs, we need design a feasible dimension-reduction fashion, thus, the Heterogeneous Mean-Field (HMF) theory is used to approximate the entire graph’s perturbation into one-dimensional function (Eq. 6).
>
> **Step IV: Exploring ASEP**
>
> Given the condensed one-dimensional function, we need utilize the topology information (degree centrality) to define some adapted parameters ($H$ and $a$ in Eq. 9) to satisfy two conditions: i) well-match adversarial perturbations (Figs. 2-6); and ii) warrant the existence of ASEP, i.e. adhere to Theorem 2.6.
>
> **Why Degree Centrality Is Valid and Sufficient.**
> We explain the behind reasons from the viewpoint of theory. As well known, three basic factors reflect the topology property: degree centrality, betweenness, and closeness. Nevertheless, our method in principle inspects the graph robustness from the angle of graph topology, and from which the ASEP can be inferred in theory. That is to say, no matter from degree centrality, betweenness, or closeness, the robust structure can be finally obtained under the guidance of ASEP. In fact, before manuscript submission, we have employed the Triple (degree-centrality + betweenness + closeness) to run experiments, and the similar results are obtained.
>
> The validity and sufficiency can be evidenced by Figs. 2-6, where we can observe the degree centrality-based resilience curves well-match the different-level adversarial perturbations.

---

> ### Author Response · Authors · 2025-11-22
> **Reply to Q2**
>
> **Q2.** Why were more recent, state-of-the-art defense methods such as Pro-GNN not included in the comparisons? How does EquiliRes perform in direct comparison to these methods?
>
> **Reply:** We understand the importance of key recent robust GNN methods and representative adversarial attacks.
>
> In current version, not only are the preprocessing methods used as baselines, such as GCN (Kipf & Welling, 2017), GAT (Velickovic et al., 2018), RGCN (Zhu et al., 2019), GCN-SVD (Entezari et al., 2020), Mid-GCN (Huang et al., 2023), but the graph neural flow and adversarial training methods are also involved, such as HANG (Zhao et al., 2023), GOOD-AT (Li et al., 2024). Simultaneously, we also utilize three non-targeted attacks: Metattack (Zugner & Gunnemann, 2019), CE-PGD (Xu et al., 2019), DICE (Waniek et al., 2016), and one recently-proposed targeted attack: GOttack (Alom et al., 2025).
>
> Different from our method from the viewpoint of preprocessing, Pro-GNN is model-retraining method, it leverages the deep learning technique to iteratively optimize the adjacency-matrix to enhance the robustness. As shown in in Fig. 12., it is time-consuming.
>
> To verify the effectiveness of our method prehensively, we additionally use the newest targeted attack GOttack (Alom et al., 2025) to run another group of experiments with the comparison to baselines GCN, GCD-SVD, Mid-GCN, Pro-GNN. Seen from Table R8, we can see our method can significantly promote the robustness of the existing baselines.
>
> Table R8: Accuracy of node classification under GOttack (Targeted-Attack Node (TAN) No.: 40/80, budget=1).
>
> |Datasets|TAN|GCN|GCD-SVD|Ours+GCN-SVD|Mid-GCN|Ours+Mid-GCN|Pro-GNN|Ours+Pro-GNN|
> |-|-|-|-|-|-|-|-|-|
> |Citeseer|40|0.4650±0.0515|0.6150±0.0374|**0.6450±0.0797**|0.6500±0.0447|**0.7100±0.0406**|0.5800±0.0781|**0.6500±0.0894**|
> ||80|0.4300±0.0170|0.6125±0.0474|**0.6575±0.0727**|0.6950±0.0584|**0.7150±0.0348**|0.5450±0.0655|**0.5850±0.0414**|
> |Cora|40|0.5650±0.0682|0.6300±0.0187|**0.6650±0.1125**|0.7350±0.0718|**0.7950±0.0797**|0.8050±0.0660|**0.8400±0.0200**|
> ||80|0.5400±0.0406|0.6600±0.0366|**0.7325±0.0491**|0.6675±0.0444|**0.6975±0.0366**|0.7875±0.0440|**0.7975±0.0242**|
> |polblogs|40|0.5900±0.0255|0.9000±0.0224|**0.9050±0.0292**|0.8800±0.0332|**0.9250±0.0274**|0.7800±0.0678|**0.8200±0.0534**|
> ||80|0.6925±0.0451|0.9475±0.0184|**0.9525±0.0146**|0.9425±0.0170|**0.9725±0.0278**|0.8700±0.0232|**0.8900±0.0374**|
>
> Besides the experiments above, we also directly compare our method to Pro-GNN on dataset Polblogs under the three non-targeted attacks. From the results in Table R9, we know that our EquiliRes performs better than Pro-GNN.
>
> Table R9: Accuracy of node classification on Polblogs.
>
> |GAA|Per.| 5%|10%|15%|20%|25%|
> |-|-|-|-|-|-|-|
> |Metattack|Pro-GNN|0.8854±0.0262|0.8000±0.0269|0.7379±0.0121|0.7011±0.0061|0.6811±0.0107|
> ||EquiliRes|**0.9011±0.0262**|**0.8970±0.0222**|**0.8653±0.0147**|**0.8141±0.0194**|**0.7311±0.0260**|
> |CE-PGD|Pro-GNN|0.8403±0.0069|0.7926±0.0082|0.7689±0.0082|0.7530±0.0070|0.7416±0.0098|
> ||EquiliRes|**0.8669±0.0119**|**0.8267±0.0112**|**0.7993±0.0139**|**0.7682±0.0101**|**0.7559±0.0141**|
> |DICE|Pro-GNN|0.9017±0.0126|0.8681±0.0147|0.8430±0.0075|0.8231±0.0095|0.8002±0.0060|
> ||EquiliRes|**0.8947±0.0095**|**0.8843±0.0232**|**0.8685±0.0211**|**0.8475±0.0154**|**0.8345±0.0163**|

---

> ### Author Response · Authors · 2025-11-22
> **Reply to Q3**
>
> **Q3.** The method focuses on purifying graph topology. Does the theoretical framework provide any insight into defending against perturbations to node features as well?
>
> **Reply:** Despite our method only focuses on the graph topology at present, our proposed theoretical framework are in fact applicable to N-dimensional space. To verify that, we produce a set of new experiments to show the performance regarding 2D modeling on both topology and features (EquiliRes(Top.+Fea.)), i.e. defending against perturbations to both graph topology and node features.
>
> From the experimental results in Table R10, we can observe EquiliRes(Top.+Fea.) significantly enhances the robustness compared to the analogous preprocessing method GCN-SVD and GCN-Jaccard.  Similar results are gained on dataset Cora_ML in Table R3 (Reply to Reviewer 1DPw).
>
> Furthermore, we combine ResiliRes with GCN, GAT, HANG, and Mid-GCN, and compare to their original version under Metattack attack, that is to say, we in advance modify the adjacency matrix referring to ASEP, then perform the baselines. As shown in Table R11, the experimental results show our method can significantly boost the adversarial resilience of these neural network-optimized baselines. Similar results are obtained on dataset Cora in Table R4 (Reply to Reviewer 1DPw).
>
> Table R10: Accuracy of node classification on Citeseer.
>
> |GAA |Per.|0%|5%|10%|15%|20%|25%|
> |-|-|-|-|-|-|-|-|
> |Metattack|GCN-SVD|0.6831±0.0126|0.6686±0.0149|0.6456±0.0216|0.5634±0.1718|0.5832±0.0254|0.5708±0.0285|
> ||GCN-Jaccard|0.7264±0.0126|0.7098±0.0148|0.6802±0.0197|0.6469±0.0264|0.6227±0.0161|0.5956±0.0310|
> ||EquiliRes(Top.+Fea.)|**0.7264±0.0135**|**0.7159±0.0170**|**0.8219±0.0114**|**0.6885±0.0170**|**0.6754±0.0147**|**0.6647±0.0189**|
> |CE-PGD|GCN-SVD|0.6831±0.0126|0.6773±0.0106|0.6763±0.0127|0.6960±0.0067|0.6875±0.0142|0.6842±0.0119|
> ||GCN-Jaccard|0.7264±0.0126|0.7184±0.0140|0.7167±0.7167|0.7137±0.0128|0.7136±0.0134|0.7111±0.0144|
> || EquiliRes(Top.+Fea.)|**0.7264±0.0135**|**0.7226±0.0137**|**0.7226±0.0114**|**0.7221±0.0113**|**0.7185±0.0111**|**0.7146±0.0111**|
> |DICE|GCN-SVD|0.6831±0.0126|0.6555±0.0161|0.6461±0.0230|0.6232±0.0155|0.6165±0.0178|0.5978±0.0213|
> ||GCN-Jaccard|0.7264±0.0126|0.7185±0.0136|0.7081±0.0123|0.6964±0.0141|0.6834±0.0137|0.6777±0.0123|
> ||EquiliRes(Top.+Fea.)|**0.7264±0.0135**|**0.7126±0.0156**|**0.7116±0.0141**|**0.7008±0.0134**|**0.6964±0.0179**|**0.6847±0.0128**|
>
> Table R11: Accuracy of node classification on Amazon Photo under combination with Ours using Metattack.
>
> |Per.|0%|5%|10%|15%|20%|25%|
> |-|-|-|-|-|-|-|
> |GCN|0.9215±0.0034|0.8242±0.0774|0.8281±0.0581|0.7529±0.1425|0.6295±0.1496|0.6108±0.1526|
> |EquiliRes(Top.+Fea.)|**0.9243±0.0118**|**0.8486±0.0682**|**0.8532±0.0338**|**0.7754±0.1088**|**0.7108±0.1246**|**0.6847±0.1278**|
> |GAT|0.9354±0.0018|0.9173±0.0039|0.7824±0.1926|0.7239±0.1575|0.6528±0.1737|0.5811±0.1824|
> |Ours+GAT|**0.9347±0.0032**|**0.9162±0.0054**|**0.9151±0.0039**|**0.9093±0.0068**|**0.9016±0.0056**|**0.8992±0.0061**|
> |HANG|0.9347±0.0032|0.9162±0.0054|0.9151±0.0039|0.9093±0.0068|0.9016±0.0056|0.8992±0.0061|
> |Ours+HANG|**0.9441±0.0025**|**0.9245±0.0045**|**0.9236±0.0033**|**0.9203±0.0058**|**0.9184±0.0048**|**0.9052±0.0052**|
> |Mid-GCN|0.7725±0.0025|0.7439±0.0049|0.6595±0.0082|0.5708±0.0170|0.4875±0.0088|0.4675±0.0113|
> |Ours+Mid-GCN|**0.7903±0.0053**|**0.7473±0.0064**|**0.6651±0.0067**|**0.5969±0.0102**|**0.5020±0.0088**|**0.4759±0.0101**|

---

> ### Comment · Reviewer_9ezX · 2025-11-25
>
> Having reviewed the authors’ response, I confirm that my concerns have been resolved and no additional inquiries arise. I wish to keep my score.

---

### Official Review · Reviewer_niN4 · 2025-10-30

**Soundness:** 2
**Presentation:** 1
**Contribution:** 2
**Rating:** 2
**Confidence:** 3

**Summary:**

This paper proposes a novel theoretical and algorithmic framework, EquiliRes, for understanding and improving graph adversarial resilience through the concept of dynamic equilibrium. The authors model adversarial perturbations on graphs as a complex dynamic system and establish the existence of an asymptotically stable equilibrium point that represents the graph’s critical adversarial resilience state. Building upon this theoretical foundation, the paper introduces a one-dimensional perturbation mapping and an iterative procedure to construct an attack-resilient topology. Experimental results on five common benchmark datasets demonstrate promising robustness improvements compared to several baseline defenses.

**Strengths:**

* The paper presents an interesting theoretical perspective that connects adversarial graph learning with complex dynamic systems.

* The equilibrium-based framework offers a potentially generalizable and interpretable way to reason about adversarial robustness.

* Experimental results indicate consistent performance gains across different attacks and datasets, suggesting the practical potential of the proposed approach.

**Weaknesses:**

* Insufficient baselines:
The comparison is incomplete. Key recent robust GNN methods, such as RUNG [1], are missing from the evaluation. Including these would make the empirical validation more convincing.

[1] Hou, Z., Feng, R., Derr, T., & Liu, X. (2023). Robust Graph Neural Networks via Unbiased Aggregation

* Weak experimental completeness:
The experiments are not comprehensive. For instance, in Table 2 (GOttack), only one baseline (GCN-SVD) is reported, which is far from sufficient for a fair comparison. Similarly, ablation and parameter sensitivity analyses could be expanded to strengthen the empirical section.

* Poor writing quality:
The overall writing, particularly in the experimental sections, is difficult to follow. Descriptions of settings, metrics, and observations are often unclear and overly verbose. The readability and logical flow of the experimental analysis need major improvement.

**Questions:**

See weaknesses.

---

> ### Author Response · Authors · 2025-11-22
> **Reply to Q1-Q2**
>
> At first, to clarify this paper’s topic and merit, we cite the summary comments by other Reviewer.
>
> **Paper’s Topic and Merit**
>
> This paper presents a novel and conceptually significant contribution to the field of adversarial machine learning on graph data. The key insight of modeling adversarial perturbations as oscillations in a complex dynamic system (CDS) and identifying the critical attack-resilient state via asymptotically stable equilibrium points (ASEP) represents a promising direction towards effective defense against graph adversarial attacks. This theoretical grounding not only explains why graph models are vulnerable, but also provides a principled framework from the angle of pure graph topology. This approach offers a more holistic and generalizable defense strategy.
>
> The formulation of adversarial resilience as a stability problem in CDS opens up a rich avenue for future research: one can envision follow-up work on identifying other dynamical invariants or bifurcation points that signal vulnerability, extending the framework to temporal or heterogeneous graphs, or applying similar principles to non-graph domains such as natural language or computer vision. The experimental validation across multiple datasets and attack types further strengthens the credibility of the approach and provides a solid foundation for benchmarking.
>
> I respectfully anticipate the Reviewer and Area Chair to thoroughly review our work and fully understand our paper's contribution and value, even though there are mass mathematical formulas that may influence the fluent presentation. The following are the replies to reviewer's concerns point by point.
>
> **Q1:** The comparison is incomplete. Key recent robust GNN methods, such as RUNG, are missing from the evaluation.
>
> **Reply:** We understand the importance of key recent robust methods and representative adversarial attacks. In current version, not only are the preprocessing methods used as baselines, such as GCN (Kipf & Welling, 2017), GAT (Velickovic et al., 2018), RGCN (Zhu et al., 2019), GCN-SVD (Entezari et al., 2020), Mid-GCN (Huang et al., 2023), but the graph neural flow and adversarial training methods are also involved, such as HANG (Zhao et al., 2023), GOOD-AT (Li et al., 2024).
>
> Furthermore, we also choose three non-targeted attacks: Metattack (Zugner & Gunnemann, 2019), CE-PGD (Xu et al., 2019), DICE (Waniek et al., 2016), and one recently-proposed targeted attack: GOttack (Alom et al., 2025).
>
> Thanks for recommending RUNG (Hou et al., 2023), we run experiments to compare it with ours. Seen from the Table R5, we can see our method outperforms RUNG, this is because RUNG cannot tackle those perturbed edges that connect dissimilar nodes, especially when perturbation-strength enlarges from 5% to 25%. Differently, our method always attempts to find the ASEP (resilience state) globally against whatever perturbation strengths.
>
> Table R5: Perform comparison under Metattack.
>
> |Dataset|Per.|5%|10%|15%|20%|25%|
> |-|-|-|-|-|-|-|
> |Cora_ML|RUNG|**0.6853±0.0129**|0.6300±0.0074|0.5934±0.0136|0.5644±0.011|0.5314±0.0210|
> ||EquiliRes |0.6599±0.0061|**0.6515±0.0061**|**0.6423±0.0036**|**0.6287±0.0045**|**0.6148±0.0043**|
> |Cora|RUNG|0.5600±0.0247|0.4686±0.0316|0.4375±0.0409|0.4084±0.0347|0.3690±0.0147|
> ||EquiliRes|**0.5623±0.0044**|**0.5471±0.0065**|**0.5281±0.0093**|**0.5156±0.0109**|**0.5008±0.0105**|
>
> **Q2:** The experiments are not comprehensive. In Table 2 (GOttack) in main text, only one baseline (GCN-SVD) is reported, which is far from sufficient for a fair comparison.
>
> **Reply:**
> Thanks for pointing out this. We add a group of experiments using GOttack with baselines GCN, GCN-SVD, Mid-GCN, Pro-GNN. From the results in Table R6, we observe our method can still remarkably promote the existing baselines.
>
> Table R6: Accuracy of node classification under GOttack (Targeted-Attack Node (TAN) No.: 40/80, budget=1).
>
> |Datasets|TAN|GCN|GCD-SVD|Ours+GCN-SVD|Mid-GCN|Ours+Mid-GCN|Pro-GNN|Ours+Pro-GNN|
> |-|-|-|-|-|-|-|-|-|
> |Citeseer|40|0.4650±0.0515|0.6150±0.0374|**0.6450±0.0797**|0.6500±0.0447|**0.7100±0.0406**|0.5800±0.0781|**0.6500±0.0894**|
> ||80|0.4300±0.0170|0.6125±0.0474|**0.6575±0.0727**|0.6950±0.0584|**0.7150±0.0348**|0.5450±0.0655|**0.5850±0.0414**|
> |Cora|40|0.5650±0.0682|0.6300±0.0187|**0.6650±0.1125**|0.7350±0.0718|**0.7950±0.0797**|0.8050±0.0660|**0.8400±0.0200**|
> ||80|0.5400±0.0406|0.6600±0.0366|**0.7325±0.0491**|0.6675±0.0444|**0.6975±0.0366**|0.7875±0.0440|**0.7975±0.0242**|
> |polblogs|40|0.5900±0.0255|0.9000±0.0224|**0.9050±0.0292**|0.8800±0.0332|**0.9250±0.0274**|0.7800±0.0678|**0.8200±0.0534**|
> ||80|0.6925±0.0451|0.9475±0.0184|**0.9525±0.0146**|0.9425±0.0170|**0.9725±0.0278**|0.8700±0.0232|**0.8900±0.0374**|

---

> ### Author Response · Authors · 2025-11-22
> **Reply to Q3**
>
> **Q3:** Ablation and parameter sensitivity analyses could be expanded to strengthen the empirical section.
>
> **Reply:**  We explain this concerns as follows:
>
> **(1) Ablation Studies**
>
> ASEP associates to the critical state of adversarial resilience, thus we run the ablation experiments without referring to ASEP as guidance, and compare to the current version with ASEP on three the datasets Polblogs, Cora_ML and Cora. As shown in Table 4 and Tables 10-11 in main text, we know the performance is significantly boosted by ASEP.
>
> **(2) Parameter Sensitivity**
> We extend the rate of adversarial perturbation from 30% to 45% to analyze whether equilibrium points ($\tilde \beta$, $\tilde x$) can still locate at the theoretical ASEP curve in Fig. 2(a).
>
> We present the results in Table R7. Pursuant to the coordinates, we find all the derived equilibrium points can locate at the theoretical ASEP curve appropriately, implying our designed 1D dynamic-variance function is reasonable.
>
> Table R7: Derived ASEP under Metattack on Polblogs.
> |Per. (%)|$\tilde \beta$|$\tilde x$| |Per. (%)|$\tilde \beta$|$\tilde x$|
> |-|-|-|-|-|-|-|
> |0|81.24|0.5324||20|73.36|0.4806|
> |3|79.23|0.5191||21|73.26|0.4800|
> |5|78.14|0.5120||24|72.78|0.4768|
> |6|77.68|0.5089||25|72.70|0.4763|
> |9|76.51|0.5013||27|72.31|0.4738|
> |10|76.22|0.4994||30|72.31|0.4738|
> |12|75.62|0.4954||35|71.92|0.4712|
> |15|74.78|0.4900||40|72.50|0.4750|
> |18|74.05|0.4851||45|72.48|0.4749|

---

### Official Review · Reviewer_1DPw · 2025-10-31

**Soundness:** 2
**Presentation:** 2
**Contribution:** 2
**Rating:** 2
**Confidence:** 4

**Summary:**

The paper asks if a graph itself possess an intrinsic, critical adversarial-resilience state, and if so, how do we find it and use it to build a more attack-resilient topology? It treats adversarial learning on graphs as a multi-object dynamic system. Under bounded, continuous perturbations, the graph’s state can converge to an asymptotically stable equilibrium point (ASEP), or the graph’s critical resilience state.

**Strengths:**

1. The paper introduces a complex multi-object dynamic system perspective to study adversarial robustness of graph neural networks.

2. The paper obtains an asymptotically stable equilibrium point.

**Weaknesses:**

1. The intrinsic adversarial-resilience state exists only under bounded, smooth perturbations and the paper’s modelling assumptions, which can be a significant limitation for general graph structures.

2. The use of the analogy in Fig. 1 is helpful to understand the motivation of the method. However, using the analogy to guide the method design somehow oversimplifies the problem space, e.g., the node/edge states in a graph may have the correlations or dependencies that cannot be captured by the three states in the paper.

3. Decomposing the adversarial perturbation into linear and non-linear part shares similarity to gradient-based perturbation (Wu et al., 2019). How does the decomposition differ to the graph per se methods built on gradient-based perturbation patterns?

4. The method deals with topology only and node features are not handled, therefore strong defence like pro-gnn is not compared.

**Questions:**

see weaknesses.

---

> ### Author Response · Authors · 2025-11-22
> **Reply to Q1-Q2**
>
> At first, to clarify this paper’s topic and merit, we cite the summary comments by other Reviewer.
>
> **Paper’s Topic and Merit**
>
> This paper presents a novel and conceptually significant contribution to the field of adversarial machine learning on graph data. The key insight of modeling adversarial perturbations as oscillations in a dynamic system and identifying the critical attack-resilient state via asymptotically stable equilibrium points represents a promising direction towards effective defense against graph adversarial attacks. This theoretical grounding not only explains why graph models are vulnerable, but also provides a principled framework from the angle of pure graph topology. This approach offers a more holistic and generalizable defense strategy.
>
> The formulation of adversarial resilience as a stability problem in CDS opens up a rich avenue for future research: one can envision follow-up work on identifying other dynamical invariants or bifurcation points that signal vulnerability, extending the framework to temporal or heterogeneous graphs, or applying similar principles to non-graph domains such as natural language or computer vision. The experimental validation across multiple datasets and attack types further strengthens the credibility of the approach and provides a solid foundation for benchmarking.
>
> I respectfully anticipate the Reviewer and Area Chair to thoroughly review our work and fully understand our paper's contribution and value, even though there are mass mathematical formulas that may influence the fluent presentation. The following are the replies to reviewer's concerns point by point.
>
> **Q1:** The resilience state exists only under bounded perturbations and modelling assumptions, which can be a significant limitation for general graph structures.
>
> **Reply:** In theory, the resilience state requires bounded perturbations and modeling assumptions to warrant the existence of asymptotically stable equilibrium point (ASEP), which corresponds to the critical state of adversarial resilience.
>
> We answer this concern from two perspectives:
>
> **(1) Reasonability of Modeling Assumptions**
>
> For graph data, the resilience state indeed depends on bounded perturbation, e.g., you can image if all edges are removed, the graph’s resilience cannot exist. To date, there is no metric to quantify adversarial perturbations, towards this predicament, we utilize Laplace Transform to calculate perturbations and its boundary in an adversarial learning environment, which is presented by a dynamic-variation function in our work.
>
> Furthermore, for high-dimensional graph data, we leverage Heterogeneous Mean-Field (HME) approximation theory to condense the adversarial perturbations of whole graph as a 1D function. In fact, the best way is to build one dimension for each node, which can capture the fine-grained perturbations. Nevertheless, such operation may be difficult regarding the feasibility, resulting from two main reasons: i) theoretically, there is no principled guidance on how to establish an ASEP-reserved function; and ii) for high-dimensional function, the analysis on whether it has an ASEP is extremely sophisticated due to the intrinsic correlation among different dimensions.
>
> **(2) Generalization for General Graph Structures**
>
> Regarding the generalization, we think it can be tackled by applying HME theory and self-defined dynamic-variation function.
>
> On one hand, HME theory can appropriately achieve the generalization for general graphs, since HMF resorts to the basic in/out-degree of nodes to approximate the whole graph’s information into 1D expression, thus no matter whatever type of graphs, as long as there exist nodes and edges, HMF can always be applied.
>
> On the other hand, the dynamic function (Eq. 7) can be self-defined, with only two requirements: i) well-matching the adversarial perturbation curves (Fig.2-Fig.6); and ii) guarantee the existence of ASEP. Therefore, in general such a proper function can be defined.
>
> **Q2:** Using the analogy in Fig. 1 to guide the method design oversimplifies the problem space, e.g., the node/edge states in a graph may have the correlations or dependencies that cannot be captured by the three states.
>
> **Reply:** Fig. 1 demonstrates the three finally-converged states after oscillation/perturbation, no matter how long it may undergo. Even if the process of adversarial perturbation is extremely sophisticated, e.g., the node/edge states can be correlated and dependent mutually, it will converge into one of the three states pursuant to Lyapunov-Stability Theory. Thereby, the complex perturbations over nodes/edges can be captured, and eventualy fall into one state.

---

> ### Author Response · Authors · 2025-11-22
> **Reply to Q3-Q4**
>
> **Q3:** Decomposing the adversarial perturbation into linear and non-linear part shares similarity to gradient-based perturbation. How does the decomposition differ to the graph per se methods built on gradient-based perturbation patterns?
>
> **Reply:** First, we want to mention that the original motivation of our work is to propose a fundamental conjecture for graph robustness and figure out a feasible (lightweight) solution for high-dimensional graph data. To this end, we provide a rigorous theoretical framework to locate the ASEP without executing any gradient-based deep-learning operations. Then, taking ASEP as indicator, the contaminated graph can be purified into a robust state through modifying edges.
>
> To avoid the time-consuming computation in forward/backward propagation with iteratively-updated gradient information, we model the perturbation process as a linear and nonlinear function (Eq. 7), upon which, we further theoretically infer the ASEP trajectory (Figs. 2-6) without really executing forward/backward propagation. In view of this, the introduction of linear and nonlinear just servers as an auxiliary to infer ASEP, with the requirement that the ASEP trajectory ought to properly reflect the adversarial perturbations. To guarantee that, following the previous studies (Kundu et al., 2022; Gao et al., 2016), we know that the dynamics (perturbations) can be divided into linear action and nonlinear action. Of course, this perturbation-reflected function can be alternatively inferred by the gradient-based methods (like Wu et al., 2019) with extensive computation.
>
> **Q4:** Compare strong defense like Pro-GNN.
>
> **Reply:** We compare ours to Pro-GNN, which leverages deep-learning technique to iteratively optimize adjacency-matrix to enhance the robustness, it is time-consuming as shown in Fig. 12. To verify the effectiveness comprehensively, we use the newest graph adversarial attack GOttack (Alom et al., 2025) to rate the defense capabilities of GCD-SVD, Mid-GCN, Pro-GNN as shown in Table R1. Furthermore, we also run experiments to directly compare Pro-GNN with EquiliRes as shown in Table R2. The experimental results show that our method has better performance.
>
> Table R1: Accuracy of node classification under GOttack (Targeted-Attack Node (TAN) No.: 40/80, budget=1).
>
> |Datasets|TAN|GCN|GCD-SVD|Ours+GCN-SVD|Mid-GCN|Ours+Mid-GCN|Pro-GNN|Ours+Pro-GNN|
> |-|-|-|-|-|-|-|-|-|
> |Citeseer|40|0.4650±0.0515|0.6150±0.0374|**0.6450±0.0797**|0.6500±0.0447|**0.7100±0.0406**|0.5800±0.0781|**0.6500±0.0894**|
> ||80|0.4300±0.0170|0.6125±0.0474|**0.6575±0.0727**|0.6950±0.0584|**0.7150±0.0348**|0.5450±0.0655|**0.5850±0.0414**|
> |Cora|40|0.5650±0.0682|0.6300±0.0187|**0.6650±0.1125**|0.7350±0.0718|**0.7950±0.0797**|0.8050±0.0660|**0.8400±0.0200**|
> ||80|0.5400±0.0406|0.6600±0.0366|**0.7325±0.0491**|0.6675±0.0444|**0.6975±0.0366**|0.7875±0.0440|**0.7975±0.0242**|
> |polblogs|40|0.5900±0.0255|0.9000±0.0224|**0.9050±0.0292**|0.8800±0.0332|**0.9250±0.0274**|0.7800±0.0678|**0.8200±0.0534**|
> ||80|0.6925±0.0451|0.9475±0.0184|**0.9525±0.0146**|0.9425±0.0170|**0.9725±0.0278**|0.8700±0.0232|**0.8900±0.0374**|
>
> Table R2: Accuracy of node classification on Polblogs.
> |GAA |Per. |5%|10%|15%|20%|25%|
> |-|-|-|-|-|-|-|
> |Metattack|Pro-GNN|0.8854±0.0262|0.8000±0.0269|0.7379±0.0121|0.7011±0.0061|0.6811±0.0107|
> ||EquiliRes|**0.9011±0.0262** |**0.8970±0.0222**|**0.8653±0.0147**|**0.8141±0.0194**|**0.7311±0.0260**|
> |CE-PGD|Pro-GNN|0.8403±0.0069|0.7926±0.0082|0.7689±0.0082|0.7530±0.0070|0.7416±0.0098|
> ||EquiliRes|**0.8669±0.0119**|**0.8267±0.0112**|**0.7993±0.0139**|**0.7682±0.0101**|**0.7559±0.0141**|
> |DICE| Pro-GNN|0.9017±0.0126|0.8681±0.0147|0.8430±0.0075|0.8231±0.0095|0.8002±0.0060|
> ||EquiliRes |**0.8947±0.0095**|**0.8843±0.0232**|**0.8685±0.0211**|**0.8475±0.0154**|**0.8345±0.0163**|

---

> ### Author Response · Authors · 2025-11-22
> **Reply to Q5**
>
> **Q5:** The method deals with topology only and node features are not handled.
>
> **Reply:** At first, we want to emphasize that although our method currently focuses on topology only, our proposed theoretical framework are applicable to N-dimensional space. We produce several groups of experiments to verify this point regarding 2D modeling on both graph topology and node features (EquiliRes(Top.+Fea.)).
>
> Prior to analyzing the performance, we want to state that our method has two-aspect effects: i) compared to the preprocessing methods, our work provides a generalized and complete theoretical framework for the ASEP existence, and referring to this ASEP, it can directly enhance the graph’s robustness; and ii) compared to the neural network-optimization methods, our work can serve as a booster to promote the robustness through preliminarily tailoring the contaminated graph into an equilibrium state.
>
> From the experimental results in Table R3, we can observe EquiliRes(Top.+Fea.) significantly enhances the robustness compared to the analogous preprocessing method GCN-SVD and GCN-Jaccard. For instance, at RAP 25\%, it outperforms the second-best baseline by 16.41\%, 1.77\%, and 3.26\%, respectively. Similar results are gained on dataset Citeseer in Table R10 (Reply to Reviewer 9ezX).
>
> Furthermore, we combine our ResiliRes with GCN, GAT (Ours+GAT), HANG (Ours+HANG), and Mid-GCN ((Ours+Mid-GCN), and compare to their original version under Metattack attack, that is to say, we in advance modify the adjacency matrix referring to ASEP, then perform the baselines. As shown in Table R4, the experimental results show our method can significantly boost the adversarial resilience of these neural network-optimized baselines. Similar results are obtained on dataset Amazon Photo in Table R11 (Reply to Reviewer 9ezX).
>
> Table R3: Accuracy of node classification on Cora_ML.
>
> |GAA|Per.|0%|5%|10%|15%|20%|25%|
> |-|-|-|-|-|-|-|-|
> |Metattack|GCN-SVD|0.8271±0.0080|0.8164±0.0084|0.8054±0.0092|0.7857±0.0095|0.6967±0.1935|0.6279±0.1729|
> ||GCN-Jaccard|0.8423±0.0094|0.7896±0.0087|0.7441±0.0123|0.7028±0.0145|0.6552±0.0294|0.5923±0.0471|
> ||EquiliRes(Top.+Fea.)|**0.8524±0.0096**|**0.8524±0.0096**|**0.8219±0.0114**|**0.8150±0.0107**|**0.8086±0.0104**|**0.7920±0.0128**|
> |CE-PGD|GCN-SVD|0.8271±0.0080|0.8229±0.0066|0.8164±0.0080|0.8118±0.0122|0.8110±0.0089|0.8080±0.0055|
> ||GCN-Jaccard|0.8423±0.0094|0.8334±0.0106|0.8248±0.0112|0.8153±0.0086|0.8131±0.0109|0.8117±0.0082|
> ||EquiliRes(Top.+Fea.)|**0.8524±0.0096**|**0.8357±0.0092**|**0.8327±0.0112**|**0.8307±0.0086**|**0.8305±0.0092**|**0.8294±0.0090**|
> |DICE|GCN-SVD|0.8271±0.0080|0.8148±0.0065|0.8000±0.0064|0.7901±0.0089|0.7730±0.0119|0.7590±0.0087|
> ||GCN-Jaccard|0.8423±0.0094|0.8300±0.0092|0.8177±0.0101|0.8061±0.0088|0.7915±0.0135|0.7752±0.0128|
> ||EquiliRes(Top.+Fea.)|**0.8524±0.0096**|**0.8354±0.0089**|**0.8291±0.0113**|**0.8242±0.0103**|**0.8160±0.0122**|**0.8078±0.0079**|
>
> Table R4: Accuracy of node classification on Cora under Metattack.
>
> |Per.|0%|5%|10%|15%|20%|25%|
> |-|-|-|-|-|-|-|
> |GCN|0.8300±0.0100|0.7535±0.0184|0.6683±0.0340|0.5908±0.0408|0.5055±0.0330|0.4397±0.0358|
> |EquiliRes(Top.+Fea.) |**0.8324±0.0101**|**0.8032±0.0128**|**0.7886±0.0131**|**0.7807±0.0120**|**0.7709±0.0126**|**0.7521±0.0159**|
> |GAT|0.8328±0.0101|0.7965±0.0175|0.7784±0.0190|0.7136±0.0463|0.6655±0.0390|0.6178±0.0441|
> |Ours+GAT|**0.8388±0.0110**|**0.8122±0.0122**|**0.8019±0.0121**|**0.7974±0.0109**|**0.7790±0.0126**|**0.7704±0.0155**|
> |HANG|0.8192±0.0092|0.7894±0.0137|0.7724±0.0185|0.7419±0.0256|0.7154±0.0245|0.6946±0.0438|
> |Ours+HANG|**0.8346±0.0090**|**0.8036±0.0117**|**0.7867±0.0145**|**0.7852±0.0110**|**0.7731±0.0132**|**0.7671±0.0190**|
> |Mid-GCN|0.8310±0.0041|0.7581±0.1582|0.7888±0.0093|0.7543±0.0111|0.7550±0.0135|0.7443±0.0083|
> |Ours+Mid-GCN|**0.8323±0.0036**|**0.8148±0.0037**|**0.7944±0.0078**|**0.7641±0.0118**|**0.7645±0.0049**|**0.7469±0.0121**|

---

### Meta-Review · Area_Chair_eBxj · 2025-12-23

**Summary:**

This paper introduces EquiliRes, a principled framework for characterizing and enhancing adversarial robustness in graphs from a dynamical systems viewpoint.  By interpreting adversarial perturbations as the evolution of a complex dynamic process, the authors identify a critical resilience regime of graph structures and theoretically establish the presence of an asymptotically stable equilibrium.  Based on this insight, an efficient perturbation mapping and an iterative topology refinement mechanism are proposed to guide the construction of more attack-resilient graphs.  Empirical evaluations on five standard benchmark datasets validate the effectiveness of the proposed approach.
The reviewers raised questions about the theoretical assumptions, methodological details and experimental design of the paper, and expressed the rejection evaluation. Although the authors attempted to reply, the statement might not have convinced the reviewers.

**Reviewer Concerns:**

The author has put forward some intuitive responses to the theoretical assumptions. However, the lack of rigorous mathematical derivation and analysis may be unacceptable to the reviewers. As for the supplementary experiments, I think it is convincing, which may address the reviewers' concerns.

**Reviewer Scores:**

For reviewer 1DPw and niN4, they might increase their scores or keep them the same.
For reviewer  9ezX, his score won't change.
Overall, I think that even though the score has improved, the reviewers' evaluations remain negative. Therefore, I tend to reject this paper.

---

### Decision · Program_Chairs · 2026-01-26

Reject